# What If We Let Forecasting Forget? A Sparse Bottleneck for Cross-Variable Dependencies

Fan Zhang [1]   Shiming Fan [1]   Hua Wang [2]

## Abstract

Multivariate time series forecasting is critical in many real-world systems, and thus modeling cross-channel dependencies is essential. Although existing methods improve overall accuracy by enhancing representations and cross-channel interactions, it remains challenging to reliably capture inter-variable dependencies under specific conditions. We observe that dependencies in real data are often state-dependent and noisy; in such cases, dense interactions can amplify spurious correlations and lead to representation over-smoothing, which may yield unreliable predictions in certain scenarios. Motivated by this, we propose MS-FLOW, a sparse-bottleneck framework that explicitly models inter-variable interaction as capacity-limited information flow. Specifically, MS-FLOW replaces fully connected communication with selective sparse routing, retaining only a few critical dependency paths and injecting cross-variable signals under a strict communication budget, thereby suppressing redundant connections and spurious-correlation propagation. Extensive experiments demonstrate that MS-FLOW learns more reliable multivariate correlations, achieving state-of-the-art forecasting accuracy on 12 real-world benchmarks while producing fewer yet more reliable dependencies, shifting multivariate forecasting from "more interaction" to "more effective interaction".

## 1. Introduction

Time series forecasting is widely used in real-world critical systems, such as power load management (Zhang et al., 2026a; Fan et al., 2025a; Ge et al., 2025a), urban traffic forecasting (Zhang et al., 2026a; Fan et al., 2025b), weather monitoring (Zhou et al., 2022; Wang et al., 2023; Zhang et al., 2026b), and financial risk control (Idrees et al., 2019). With the proliferation of sensors and information systems, many applications naturally form multivariate time series (MTS), where a single system is observed through multiple correlated variables that influence each other (Liu et al., 2024; Lin et al., 2025; Li et al., 2025b;a). This shifts forecasting from single-series modeling to learning both temporal dynamics and cross-variable dependencies.

However, we argue that the core challenge of multivariate forecasting is not simply having "more variables," but that cross-variable interactions are often noisy, time-varying, and state-dependent. On the one hand, correlation structures in real-world systems frequently drift with time and operating conditions: as illustrated in Fig. 1(a), the same pair of variables can exhibit markedly different coupling strengths across scenarios. On the other hand, observation noise and local perturbations are ubiquitous, which can easily lead models to learn apparently meaningful yet poorly generalizable spurious correlations. As shown in Fig. 1(b), local disturbances may trigger short-lived resonance, misleading the model into treating it as a stable dependency.

Existing frameworks typically fall into two strategies. Channel-independent methods (Nie et al., 2022; Tang & Zhang, 2025; Ma et al., 2026) stabilize training by weakening cross-variable coupling, but may discard useful complementary information. In contrast, dense interaction mechanisms such as attention or channel mixing (Zhang & Yan, 2023; Wu et al., 2021) allow unconstrained communication among all variables, which can propagate redundant or spurious dependencies and lead to feature diffusion and oversmoothing (Zhang et al., 2024; Hu et al., 2025) (Fig. 1(c)). Therefore, the bottleneck is often not "too little interaction", but "too much interaction".

Based on these observations, we put forward a view that runs counter to the conventional wisdom yet better aligns with the essence of forecasting: in multivariate forecasting, the key is not to build ever more complex interaction modules, but to impose an appropriate capacity constraint on cross-variable information flow, forcing the model to forget redundant connections and transmit information only

[1]Shandong Technology and Business University, Yantai, Shandong, China [2]Ludong University, Yantai, Shandong, China. Correspondence to: Hua Wang <hua.wang@ldu.edu.cn>.

*Proceedings of the 43rd International Conference on Machine Learning*, Seoul, South Korea. PMLR 306, 2026. Copyright 2026 by the author(s).

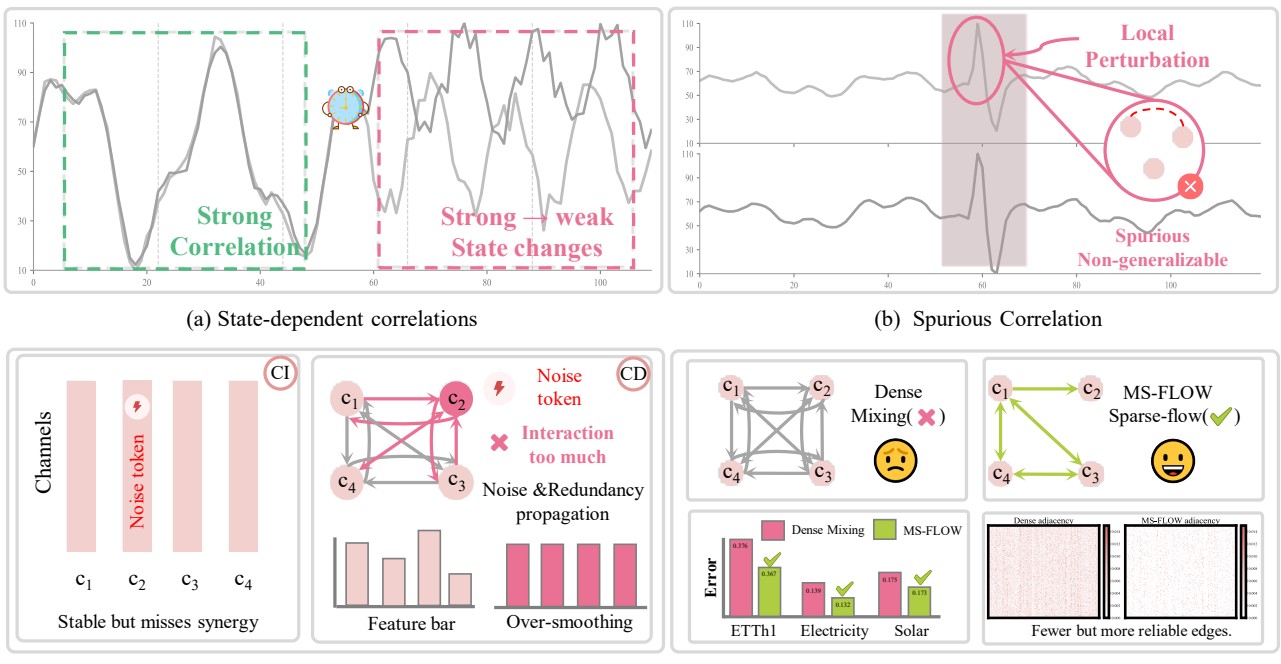

*Figure 1.* MS-FLOW's motivation and overview. (a) State dependency: Cross-variable dependencies vary over time and can switch across operating conditions. (b) Spurious correlation: Local perturbations may induce non-generalizable, misleading correlations. (c) Channel-processing strategies: Different paradigms for handling cross-channel interactions. (d) MS-FLOW: Sparse-flow bottleneck: MS-FLOW constrains cross-variable information flow via sparse routing, yielding fewer yet more reliable dependency edges and improving forecasting accuracy.

along a few critical dependency paths. To this end, we propose MS-FLOW (Multivariate Sparse-Flow), a sparse-bottleneck framework for modeling cross-variable information flow. MS-FLOW constructs a bottleneck via sparse dependencies, transforming "fully connected diffusion" into "selective routing," thereby reducing noise diffusion and spurious-correlation propagation from a mechanistic perspective. As illustrated in Fig. 1(d), MS-FLOW injects information through fewer yet more reliable dependency edges, leading to more robust forecasting performance.

Overall, MS-FLOW consists of three components. First, Temporal Patch Encoding segments the input sequence into overlapping patch tokens and maps them into patch-level representations, providing stable segment-wise features for subsequent modeling. Second, Patch-wise Temporal Interaction mixes the token sequence of each variable along the patch dimension to learn within-variable temporal relationships. Finally, Sparse Dependency Bottleneck learns sample-adaptive dependency structures from variable-level states and constructs a cross-variable information-flow bottleneck through sparsification, enabling cross-variable information injection under a limited communication budget. Since the core of MS-FLOW lies in imposing a capacity constraint on cross-variable information flow rather than simply stacking complex modules, it achieves strong fore-

casting performance with a relatively small parameter budget. Moreover, the learned sparse dependency graph offers good interpretability, providing intuitive and verifiable structural evidence of cross-variable couplings. The main contributions of this paper can be summarized as follows:

❶ **Conceptual Perspective.** We identify over-interaction as a key failure mode in multivariate forecasting: dense cross-variable mixing tends to diffuse noise and amplify spurious correlations, motivating a controlled communication principle.

❷ **Methodology.** We propose MS-FLOW, which learns sample-adaptive sparse dependency graphs and performs selective cross-variable routing under a strict communication budget, preserving only a few informative coupling paths while suppressing irrelevant variables.

❸ **Overall performance.** We validate the effectiveness of MS-FLOW on multiple real-world benchmarks. Specifically, compared with 14 state-of-the-art baselines, MS-FLOW achieves the top-1 rank on 22 out of 24 averaged metrics (MSE and MAE) across 12 standard-scale datasets.

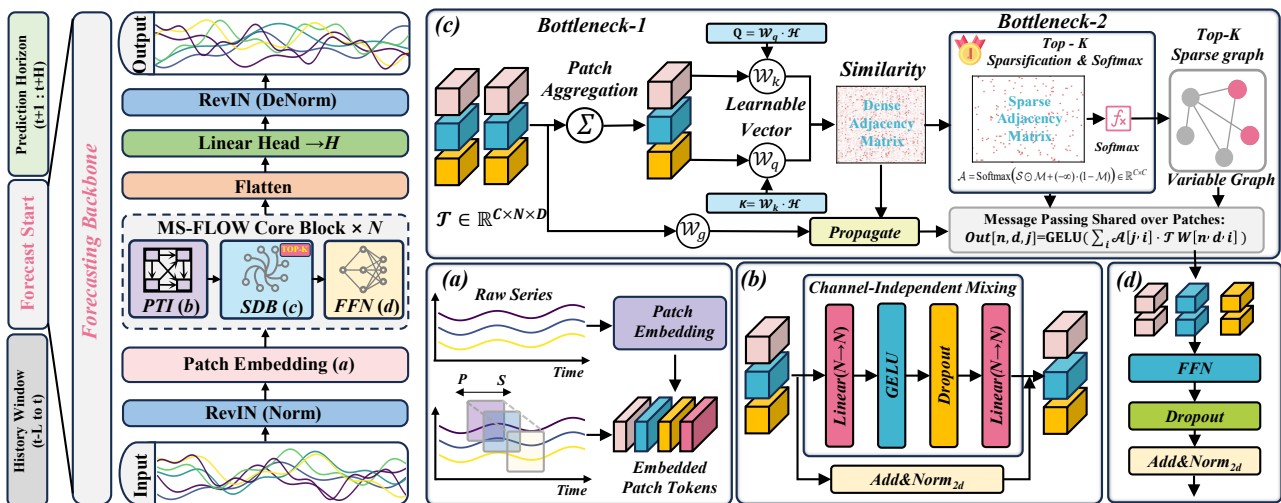

*Figure 2.* The overall architecture of MS-FLOW consists of: (*a*) Patch Embedding to represent the input X; (*b*) Patch-wise temporal interaction to capture temporal patterns; (*c*) A sparse dependency bottleneck to selectively route cross-variable information for predicting $\mathcal{Y}$.

## 2. Related Work

With the rapid development of deep learning (Yang et al., 2026; Ge et al., 2025b), modeling complex dependencies in high-dimensional data has become a central theme across many domains. In fields such as image recognition (Li et al., 2025c; 2026a; Chen et al., 2025a), video understanding (Hu et al., 2026; Chen et al., 2025b), and multimodal learning, modern neural architectures have achieved remarkable success by leveraging rich interactions among spatial, temporal, and semantic units. In multivariate time series forecasting (MTSF), how to effectively model cross-variable dependencies has long been a central challenge. Existing methods can be broadly grouped into several categories. The first category follows the channel-independent (CI) paradigm (Zeng et al., 2023; Nie et al., 2022; Ma et al., 2025; 2026; Masserano et al., 2024; Lee et al., 2025; Liu et al., 2026; Li et al., 2026b; Wang et al., 2026b;a), modeling temporal dependencies within each variable only and approximating multivariate forecasting as a collection of univariate tasks. Such methods are typically stable during training, but they almost completely discard potentially complementary information across variables, which often leads to insufficient information utilization in strongly correlated settings (Han et al., 2024; Wang et al., 2024; Qiu et al., 2026). The second category adopts the channel-dependent (CD) strategy (Yang et al., 2024; Lin et al., 2025; Liu et al., 2024; Yu et al., 2024; Zhang & Yan, 2023), modeling interactions among all variables through attention mechanisms or channel mixing to gain richer information. However, in real-world scenarios, cross-variable relations are often sparse, time-varying, and noisy: although dense interactions provide broader coverage, they may also propagate numerous weak or even spurious

correlations, causing redundancy diffusion and weakening generalization. Subsequently, some studies cluster variables based on similarity (Gustafson et al., 2013; Chen et al., 2024; Qiu et al., 2025b;c), strengthening interactions within clusters while weakening those across clusters to reduce the influence of invalid dependencies; yet such approaches usually rely on coarse-grained assignments and struggle to capture the continuous drift and fine-grained evolution of dependencies under different states, potentially missing critical relations or retaining ineffective ones. Further work introduces more sophisticated filtering and routing mechanisms (Hu et al., 2025; Zhang et al., 2024; Luo et al., 2025) at the block level, attempting to "filter out irrelevant information" by selecting and filtering spatiotemporal blocks.

In this paper, MS-FLOW views cross-variable modeling as a capacity-limited communication problem. Instead of dense diffusion followed by filtering, it enforces a sparse routing bottleneck so that information flows only through a few key dependency paths, improving robustness to redundant and spurious interactions.

## 3. Methodology

We study multivariate time series forecasting. Given a historical window $\mathcal{X} \in \mathbb{R}^{C \times L}$, where $L$ denotes the input length and $C$ denotes the number of variables, the goal is to predict the next $H$ steps:

$$\hat{\mathcal{Y}} = \Phi(\mathcal{X}) \in \mathbb{R}^{C \times H}. \tag{1}$$

To address this task, we propose **MS-FLOW**, a sparse-bottleneck framework for modeling cross-variable information flow. The overall architecture is illustrated in Fig. 2.

The pseudocode and additional implementation details are provided in Appendix E.

## 3.1. Overview of MS-FLOW

MS-FLOW consists of three key components: (i) ***Temporal Patch Encoding*** segments the input series into overlapping patch tokens and maps them into patch-level representations; (ii) ***Patch-wise Temporal Interaction*** mixes the token sequence of each variable along the patch dimension to capture local-to-mid-range temporal dependencies; (iii) ***Sparse Dependency Bottleneck*** learns sample-adaptive dependency structures from variable-level states and imposes a Top-$K$ sparsification to form a cross-variable information-flow bottleneck. Under this sparse dependency constraint, it injects cross-variable information and further projects the fused representations to produce $H$-step forecasts. Details of each component are presented in the following subsections. Our code is available at https://github.com/Cola-Fsm/MS-FLOW.

## 3.2. Temporal Patch Encoding

At this stage, we partition each variable of the input window $\mathcal{X} \in \mathbb{R}^{C \times L}$ along the time axis into a set of local patches, and map each patch into an embedded patch token (Nie et al., 2022). Since the same encoding process is applied to all variables, we first describe the procedure on a single-variable sequence $\mathbf{x} \in \mathbb{R}^L$ and then restore the variable dimension. Formally, given patch length $P$ and stride $S$, we perform patching on $\mathbf{x}$ as

$$\{\mathbf{p}_1, \ldots, \mathbf{p}_N\} = \text{Patch}(\mathbf{x}; P, S), \qquad (2)$$

where each patch $\mathbf{p}_i \in \mathbb{R}^P$, and the number of patches is $N = \left\lfloor \frac{L + \text{pad} - P}{S} \right\rfloor + 1$. Padding is applied at the end of the sequence to ensure full coverage of the input window and alignment with the stride. Next, we use a learnable mapping to project each patch from its original length $P$ to the hidden dimension $D$:

$$\mathbf{z}_i = \text{Embed}(\mathbf{p}_i) \in \mathbb{R}^D, \qquad i = 1, \ldots, N. \qquad (3)$$

In implementation, $\text{Embed}(\cdot)$ is realized by a 1D convolution with kernel size $P$ and stride $S$, which is equivalent to applying a shared linear projection to each patch, yielding the embeddings for all patches in a single pass.

Extending the above process to all variables, we obtain the patch-level representation $\mathcal{Z} \in \mathbb{R}^{C \times N \times D}$, where each variable contains $N$ patch tokens.

## 3.3. Patch-wise Temporal Interaction

Given the embedded patch-level representation $\mathcal{Z} \in \mathbb{R}^{C \times N \times D}$, Patch-wise Temporal Interaction (PTI) aims to mix the token sequence of each variable along the patch dimension $N$.

**Motivation.** We operate on the patch dimension rather than directly modeling raw timestamps, because multivariate time series often suffer from short-term noise, sudden spikes, and local disturbances, which can significantly amplify the instability of point-wise interactions. Inspired by CMoS (Si et al., 2025), patch tokens serve as segment-level abstractions that are inherently more robust to local perturbations. Learning temporal correlations at the patch scale thus better resists interference and yields more stable temporal representations. Therefore, PTI is placed before cross-variable dependency learning to ensure that subsequent structure learning is built upon more reliable temporal features.

**Technical details.** For an arbitrary variable $c$, its patch sequence is denoted as $\mathcal{Z}^{(c)} \in \mathbb{R}^{N \times D}$. We apply a two-layer perceptron along the patch dimension independently for each feature channel. Specifically, for each hidden dimension $d$, we define $\mathcal{Z}_{c,d} \in \mathbb{R}^N$, and the patch-wise mixing is formulated as

$$\text{PTI}(\mathcal{Z}_{c,d}) = \mathcal{W}_2\, \phi(\mathcal{W}_1 \mathcal{Z}_{c,d}), \qquad \mathcal{W}_1, \mathcal{W}_2 \in \mathbb{R}^{N \times N} \tag{4}$$

where $\phi(\cdot)$ denotes a nonlinear activation function. Applying this operation in parallel over all $c$ and $d$ yields

$$\tilde{\mathcal{T}} = \text{PTI}(\mathcal{Z}), \qquad \tilde{\mathcal{T}} \in \mathbb{R}^{C \times N \times D}. \tag{5}$$

We then use residual updating and normalization to obtain the temporally mixed representation:

$$\mathcal{T} = \text{Norm}_1(\mathcal{Z} + \tilde{\mathcal{T}}), \qquad \mathcal{T} \in \mathbb{R}^{C \times N \times D}, \tag{6}$$

where $\text{Norm}_1(\cdot)$ denotes 2D batch normalization that treats the patch and feature dimensions as spatial axes.

## 3.4. Sparse Dependency Bottleneck

Our goal is to explicitly learn sample-adaptive cross-variable dependencies along the variable dimension, and to restrict cross-variable information flow to a few critical paths via Top-$K$ sparsification, thereby suppressing redundant interactions and spurious correlation propagation. Unlike dense cross-variable mixing, we model cross-variable communication as a ***bottleneck***: only a small subset of selected dependency edges are allowed to transmit information.

### 3.4.1. VARIABLE-LEVEL STATE COMPRESSION (**BOTTLENECK-1**)

Cross-variable dependencies should reflect variable couplings under the current input window state, rather than being dominated by instantaneous disturbances from a local patch. Therefore, we first aggregate the tokens of each

variable along the patch dimension to obtain variable-level states:

$$\mathcal{H} = \text{AvgPool}_N(\mathcal{T}) = \frac{1}{N} \sum_{n=1}^{N} \mathcal{T}_{:,n,:} \in \mathbb{R}^{C \times D}, \qquad (7)$$

where $T_{c,n}$ denotes the feature vector of variable $c$ at the $n$-th patch position. This compression provides a more stable basis for structure learning and reduces graph construction from patch-level degrees of freedom to window-level degrees of freedom, making dependency estimation less prone to overfitting local noise. Appendix D discusses the necessity of compression.

### 3.4.2. SPARSE DEPENDENCY GRAPH ESTIMATION (BOTTLENECK-2)

We estimate dependency strengths in a low-dimensional similarity space. We define

$$\mathcal{Q} = \mathcal{H}\mathcal{W}_q, \qquad \mathcal{K} = \mathcal{H}\mathcal{W}_k, \qquad \mathcal{W}_q, \mathcal{W}_k \in \mathbb{R}^{D \times d_k}, \tag{8}$$

where $\mathcal{Q}, \mathcal{K} \in \mathbb{R}^{C \times d_k}$, and $\mathcal{W}_q, \mathcal{W}_k$ are learnable projection matrices. The dependency score matrix is computed as

$$\mathcal{S} = \frac{\mathcal{Q}\mathcal{K}^\top}{\tau} \in \mathbb{R}^{C \times C}, \qquad (9)$$

where $\tau = \sqrt{d_k}$ is a scaling factor. In our implementation, we set $d_k = 64$. Different from dense attention, we impose an explicit capacity constraint via a sparsified attention map: for each variable, we keep only the Top-$K$ similarity logits, mask the rest to $-\infty$, and then apply Softmax so that gradients flow through the retained logits. Let $\mathcal{N}_K(c)$ denote the Top-$K$ index set for the $c$-th row, and we construct a binary mask $\mathcal{M} \in \{0,1\}^{C \times C}$ by

$$\mathcal{M}_{c,j} = \mathbb{I}\big(j \in \mathcal{N}_K(c)\big). \qquad (10)$$

The final dependency graph is obtained via masked softmax normalization:

$$\mathcal{A} = \text{Softmax}\big(\mathcal{S} \odot \mathcal{M} + (-\infty) \cdot (1 - \mathcal{M})\big) \in \mathbb{R}^{C \times C}, \ (11)$$

where the softmax is applied row-wise. This formulation has two key implications: (1) **Structural bottleneck**: the effective out-degree of each node is fixed to $K$, explicitly limiting the bandwidth of cross-variable information flow; (2) **Inductive bias**: multivariate forecasting often depends on only a few critical couplings, and the Top-$K$ constraint forces the model to select the most informative interaction paths under a limited budget.

### 3.4.3. SPARSE DEPENDENCY PROPAGATION

Given the sparse dependency graph $\mathcal{A}$, we share it across all patch positions and perform information aggregation along the variable dimension. For each patch position $n$, we define

$$\mathcal{U}_n = \mathcal{T}_{:,n}\mathcal{W}_g \in \mathbb{R}^{C \times D}, \qquad \mathcal{W}_g \in \mathbb{R}^{D \times D}, \qquad (12)$$

and conduct cross-variable propagation as

$$\tilde{\mathcal{G}}_{:,n} = GELU(\mathcal{A}\mathcal{U}_n) \in \mathbb{R}^{C \times D}, \qquad n = 1, \dots, N. \ (13)$$

Stacking all patch positions yields $\tilde{\mathcal{G}} \in \mathbb{R}^{C \times N \times D}$. We then apply residual updating and normalization:

$$\mathcal{G} = \text{Norm}_2\big(\mathcal{T} + \tilde{\mathcal{G}}\big) \in \mathbb{R}^{C \times N \times D}, \qquad (14)$$

where $\text{Norm}_2(\cdot)$ follows the same channel-wise 2D normalization scheme as $\text{Norm}_1(\cdot)$.

The combination of "shared graph + patch-wise propagation" is crucial: the shared $\mathcal{A}$ makes the learned dependency structure correspond to the window state rather than instantaneous noise, while applying propagation at each patch position enables these structural dependencies to consistently act on different local segments. Overall, we rewrite cross-variable interaction from dense mixing into a sparse routing problem: information can only flow along a few selected edges, and redundant paths are explicitly cut off, alleviating noise diffusion and representation over-smoothing.

**Prediction head.** We further transform the representation after cross-variable injection and produce the final forecast. Specifically, we first apply a point-wise feed-forward network:

$$\tilde{\mathcal{O}} = \text{FFN}(\mathcal{G}) \in \mathbb{R}^{C \times N \times D}, \qquad (15)$$

and then use residual updating with batch normalization to obtain the output representation:

$$\mathcal{O} = \text{Norm}_3\big(\mathcal{G} + \tilde{\mathcal{O}}\big) \in \mathbb{R}^{C \times N \times D}. \qquad (16)$$

where $\text{Norm}_3(\cdot)$ follows the same channel-wise 2D normalization scheme as $\text{Norm}_1(\cdot)$. Finally, we flatten along the patch and feature dimensions to obtain $\mathcal{R} = \text{vec}(\mathcal{O}) \in \mathbb{R}^{C \times ND}$, and project it to the forecasting horizon via a shared linear mapping:

$$\mathcal{Y} = \mathcal{R}\mathcal{W}_o + b_o, \qquad \mathcal{W}_o \in \mathbb{R}^{ND \times H}, \ b_o \in \mathbb{R}^H. \quad (17)$$

After applying inverse normalization (Kim et al., 2021) to $\mathcal{Y}$, we obtain the final prediction $\hat{\mathcal{Y}} \in \mathbb{R}^{C \times H}$. During training, we minimize the regression loss between $\hat{\mathcal{Y}}$ and the ground-truth future sequence.

## 4. Experimental

### 4.1. Experimental Setup

**Datasets.** We evaluate the proposed method on 12 widely used real-world datasets, including the ETT and PEMS series, as well as the Electricity, Solar, Traffic, and Weather benchmarks. These datasets vary substantially in scale, number of variables, sampling frequency, and application domains, enabling a comprehensive assessment of the model's

*Table 1.* Long-term forecasting results. The input length $L$ is 96. All results are averaged across four different prediction lengths: $H =\{96, 192, 336, 720\}$. The best result is **red**, the second best result is blue. See Table 8 for full results.

| Models | MS-FLOW (Ours) | | TimeFilter (ICML'2025) | | TimeMosaic (AAAI'2026) | | xPatch (AAAI'2025) | | DUET (KDD'2025) | | Leddam (ICML'2024) | | iTransformer (ICLR'2024) | | Crossformer (ICLR'2023) | | TimesNet (ICLR'2023) | | PatchTST (ICLR'2023) | | DLinear (AAAI'2023) | |
|---|---|---|---|---|---|---|---|---|---|---|---|---|---|---|---|---|---|---|---|---|---|---|
| Metrics | MSE | MAE | MSE | MAE | MSE | MAE | MSE | MAE | MSE | MAE | MSE | MAE | MSE | MAE | MSE | MAE | MSE | MAE | MSE | MAE | MSE | MAE |
| ETTh1 | **0.415** | **0.415** | 0.420 | 0.428 | 0.425 | 0.424 | 0.443 | 0.429 | 0.443 | 0.436 | 0.431 | 0.429 | 0.454 | 0.448 | 0.529 | 0.522 | 0.458 | 0.450 | 0.469 | 0.455 | 0.456 | 0.452 |
| ETTh2 | **0.306** | **0.355** | 0.364 | 0.397 | 0.363 | 0.387 | 0.373 | 0.395 | 0.372 | 0.397 | 0.372 | 0.398 | 0.383 | 0.407 | 0.942 | 0.683 | 0.414 | 0.427 | 0.387 | 0.407 | 0.559 | 0.515 |
| ETTm1 | **0.368** | **0.375** | 0.377 | 0.393 | 0.381 | 0.381 | 0.386 | 0.403 | 0.390 | 0.393 | 0.385 | 0.397 | 0.407 | 0.410 | 0.513 | 0.495 | 0.400 | 0.406 | 0.387 | 0.400 | 0.403 | 0.407 |
| ETTm2 | **0.264** | **0.311** | 0.272 | 0.321 | 0.273 | 0.314 | 0.279 | 0.320 | 0.280 | 0.324 | 0.280 | 0.325 | 0.288 | 0.332 | 0.757 | 0.611 | 0.291 | 0.333 | 0.281 | 0.326 | 0.350 | 0.401 |
| Electricity | **0.157** | **0.246** | 0.158 | 0.256 | 0.187 | 0.279 | 0.185 | 0.267 | 0.172 | 0.259 | 0.168 | 0.263 | 0.178 | 0.270 | 0.244 | 0.334 | 0.193 | 0.295 | 0.205 | 0.290 | 0.212 | 0.300 |
| Weather | **0.234** | **0.257** | 0.239 | 0.269 | 0.251 | 0.267 | 0.247 | 0.266 | 0.251 | 0.273 | 0.242 | 0.272 | 0.258 | 0.278 | 0.259 | 0.315 | 0.259 | 0.286 | 0.259 | 0.280 | 0.265 | 0.317 |
| Solar-Energy | **0.209** | **0.218** | 0.223 | 0.250 | 0.240 | 0.270 | 0.244 | 0.241 | 0.237 | 0.233 | 0.230 | 0.264 | 0.233 | 0.262 | 0.641 | 0.639 | 0.301 | 0.319 | 0.270 | 0.307 | 0.330 | 0.401 |
| Traffic | 0.440 | **0.258** | **0.407** | 0.268 | 0.458 | 0.283 | 0.520 | 0.335 | 0.451 | 0.268 | 0.467 | 0.294 | 0.428 | 0.282 | 0.550 | 0.304 | 0.485 | 0.297 | 0.555 | 0.361 | 0.625 | 0.383 |

generalization and robustness. Details of datasets and metrics are in Appendix A and Appendix B.

**Baselines.** To evaluate the performance of MS-FLOW, we carefully select 14 representative baseline models, including TimeFilter (Hu et al., 2025), TimeMosaic (Ding et al., 2025), xPatch (Stitsyuk & Choi, 2025), DUET (Qiu et al., 2025b), TimeEmb (Xia et al., 2026), TimeKAN (Huang et al., 2025), TimeXer (Wang et al., 2024), SOFTS (Han et al., 2024), Leddam (Yu et al., 2024), iTransformer (Liu et al., 2024), Crossformer (Zhang & Yan, 2023), TimesNet (Wu et al., 2023), PatchTST (Nie et al., 2022), and DLinear (Zeng et al., 2023).

**Implementation Details.** All experiments were implemented using PyTorch (Paszke, 2019) and performed on a single NVIDIA RTX 3090 GPU (24GB). We train the models using the Adam optimizer and evaluate performance with the two most commonly used metrics in time series forecasting, MSE and MAE. The Top-K in the Sparse Dependency Bottleneck is selected via empirical search. Additional hyperparameter settings, training details, and the loss function definition are provided in Appendix B.

### 4.2. Long-term Forecasting

**Setups.** We conduct long-term forecasting experiments on a range of widely used real-world benchmarks, including the Electricity Transformer Temperature (ETT) dataset with its four subsets (ETTh1, ETTh2, ETTm1, ETTm2), as well as the Weather, Electricity, Traffic, and Solar datasets. Following the iTransformer (Liu et al., 2024) setting, we fix the input length to $L = 96$ for all experiments. For fairness, we follow the TFB setting (Qiu et al., 2024) and do not use the trick of discarding the last batch of data.

**Results.** As shown in Table 1, MS-FLOW achieves consistent improvements over multiple strong baselines on long-term forecasting. First, compared with TimeFilter—the best-performing baseline overall and one that also empha-

sizes cross-variable modeling—MS-FLOW delivers gains of 2.72% / 5.69%, indicating that the proposed Top-K sparse information-flow bottleneck further suppresses redundant interactions and improves generalization. Second, MS-FLOW also shows clear advantages over two recent patch-based methods: it improves upon TimeMosaic by 7.18% / 6.53% and upon xPatch by 10.61% / 8.32%, suggesting that patch partitioning alone is insufficient to fully capture cross-variable relationships. Finally, compared with the widely adopted multivariate Transformer framework iTransformer, MS-FLOW achieves substantial gains of 8.98% / 9.44%, further validating that restricting cross-variable information flow and preserving key dependency paths is more effective than introducing denser interactions for multivariate forecasting. Appendix C provides further details.

### 4.3. Short-term Forecasting

**Setups.** For short-term forecasting, we evaluate on the PeMS series datasets, which record sensor observations from city-scale traffic networks and exhibit complex spatiotemporal correlations among multiple variables. Following a unified setting, we fix the input length to $L = 96$ for all methods and consider forecasting horizons $H \in \{12, 24, 48\}$.

**Results.** As shown in Table 2, channel-independent methods such as TimeMosaic and xPatch, while performing well on long-term forecasting, suffer a notable performance drop on PeMS due to the stronger cross-variable correlations. In contrast, TimeFilter and iTransformer, which explicitly model cross-variable relationships, can better exploit variable dependencies and thus achieve more competitive results. Notably, MS-FLOW attains the best performance on this task and further surpasses the latest multivariate model TimeFilter.

Additional comparison results are presented in Appendix C.

*Table 2.* Short-term forecasting results. The input length $L$ is 96. All results are averaged across three different forecasting horizons: $H = \{12, 24, 48\}$. The best result is **red**, the second best result is blue. See Table 11 for full results.

| Models | MS-FLOW (Ours) | | TimeFilter (ICML'2025) | | TimeMosaic (AAAI'2026) | | xPatch (AAAI'2025) | | DUET (KDD'2025) | | Leddam (ICML'2024) | | iTransformer (ICLR'2024) | | Crossformer (ICLR'2023) | | TimesNet (ICLR'2023) | | PatchTST (ICLR'2023) | | DLinear (AAAI'2023) | |
|---|---|---|---|---|---|---|---|---|---|---|---|---|---|---|---|---|---|---|---|---|---|---|
| Metrics | MSE | MAE | MSE | MAE | MSE | MAE | MSE | MAE | MSE | MAE | MSE | MAE | MSE | MAE | MSE | MAE | MSE | MAE | MSE | MAE | MSE | MAE |
| PEMS03 | **0.079** | **0.183** | 0.084 | 0.191 | 0.090 | 0.196 | 0.135 | 0.238 | 0.086 | 0.191 | 0.101 | 0.210 | 0.096 | 0.204 | 0.138 | 0.253 | 0.119 | 0.225 | 0.151 | 0.265 | 0.219 | 0.328 |
| PEMS04 | **0.081** | **0.184** | 0.083 | 0.186 | 0.100 | 0.205 | 0.153 | 0.261 | 0.096 | 0.203 | 0.102 | 0.213 | 0.098 | 0.207 | 0.145 | 0.267 | 0.109 | 0.220 | 0.162 | 0.273 | 0.236 | 0.350 |
| PEMS07 | **0.069** | **0.161** | 0.071 | 0.170 | 0.091 | 0.199 | 0.130 | 0.227 | 0.076 | 0.176 | 0.087 | 0.192 | 0.088 | 0.190 | 0.181 | 0.272 | 0.106 | 0.208 | 0.166 | 0.270 | 0.241 | 0.343 |
| PEMS08 | 0.088 | **0.178** | **0.083** | 0.186 | 0.110 | 0.228 | 0.159 | 0.249 | 0.096 | 0.192 | 0.102 | 0.211 | 0.127 | 0.212 | 0.232 | 0.270 | 0.150 | 0.244 | 0.238 | 0.289 | 0.281 | 0.366 |

## 4.4. Ablation Study

To verify the effectiveness of the proposed MS-FLOW, we conduct a systematic ablation study at the architectural level. In our experiments, the default setting is highlighted in pink . Specifically, we investigate the following questions:

- **RQ1:** Are the key components necessary, and does each make an independent contribution?

- **RQ2:** Is the proposed sparse bottleneck superior to dense or unstructured cross-variable interactions?

- **RQ3:** Is the bottleneck construction strategy (e.g., variable-level state aggregation and shared dependency graph) reasonable?

*Table 3.* Each variant removes one module from the full model while keeping all other settings unchanged.

| Case | Variant | Electricity | | Solar-Energy | | Weather | | PEMS04 | |
|---|---|---|---|---|---|---|---|---|---|
| | | MSE | MAE | MSE | MAE | MSE | MAE | MSE | MAE |
| ① | Full | **0.132** | **0.222** | **0.173** | **0.197** | **0.150** | **0.187** | **0.068** | **0.167** |
| ② | w/o PTI | 0.135 | 0.225 | 0.176 | 0.200 | 0.151 | 0.189 | 0.069 | 0.168 |
| ③ | w/o SDB | 0.152 | 0.237 | 0.210 | 0.219 | 0.163 | 0.199 | 0.086 | 0.189 |
| ④ | w/o FFN | 0.146 | 0.234 | 0.211 | 0.219 | 0.151 | 0.189 | 0.077 | 0.179 |

**Ablation on Core Components (RQ1).** We study three variants by removing one component at a time while keeping all other settings unchanged (Table 3). *w/o PTI* disables the patch-wise temporal interaction module, testing whether temporal dependency extraction at the patch scale is necessary before learning cross-variable relations. *w/o SDB* removes the sparse dependency bottleneck, testing whether explicit cross-variable dependency modeling is the key driver of performance. *w/o FFN* removes the token-wise feed-forward refinement, testing whether additional nonlinearity after dependency injection contributes to accuracy. Results show that all removals hurt performance, with *w/o SDB* leading to the largest degradation, demonstrating that the sparse cross-variable bottleneck is the main contributor, while PTI and FFN provide complementary gains.

**Ablation on Sparsity and Capacity Control (RQ2).** We examine whether our **Top-K** bottleneck is superior to dense

*Table 4.* Ablation on sparsification strategies in the Sparse Dependency Bottleneck.

| Case | Variant | Electricity | | ETTh1 | | ETTm2 | | Solar-Energy | |
|---|---|---|---|---|---|---|---|---|---|
| | | MSE | MAE | MSE | MAE | MSE | MAE | MSE | MAE |
| ① | Top-$K$ | **0.132** | **0.222** | **0.367** | **0.389** | **0.166** | **0.245** | **0.173** | **0.197** |
| ② | Dense | 0.139 | 0.231 | 0.376 | 0.390 | 0.166 | 0.246 | 0.175 | 0.200 |
| ③ | Random-$K$ | 0.136 | 0.226 | 0.369 | 0.392 | 0.167 | 0.247 | 0.179 | 0.202 |
| ④ | Static-$K$ | 0.139 | 0.230 | 0.375 | 0.395 | 0.171 | 0.250 | 0.196 | 0.209 |

or unstructured interaction (Table 4). *Dense (K=C)* disables **Top-K** pruning so that every variable communicates with all others, testing whether unrestricted mixing is beneficial. *Random-K* keeps the same number of edges but selects neighbors uniformly at random, testing whether sparsity alone (without meaningful structure) is sufficient. *Static-K* constructs a fixed dependency graph shared by all samples, testing whether sample-adaptive dependency estimation is necessary. **Top-K** consistently performs best, verifying that cross-variable communication should be capacity-limited, structured, and sample-adaptive, rather than dense, random, or static.

*Table 5.* Ablation on bottleneck construction strategies in Sparse Dependency Bottleneck.

| Case | Variant | Electricity | | ETTh2 | | ETTm1 | | Weather | |
|---|---|---|---|---|---|---|---|---|---|
| | | MSE | MAE | MSE | MAE | MSE | MAE | MSE | MAE |
| ① | Ours | **0.132** | **0.222** | **0.228** | **0.299** | **0.302** | **0.330** | **0.150** | **0.187** |
| ② | MaxPool | 0.151 | 0.238 | 0.235 | 0.303 | 0.315 | 0.343 | 0.172 | 0.214 |
| ③ | LastPool | 0.141 | 0.232 | 0.243 | 0.308 | 0.315 | 0.341 | 0.190 | 0.236 |
| ④ | Per-Patch Graph | 0.134 | 0.225 | 0.237 | 0.304 | 0.312 | 0.340 | 0.155 | 0.196 |

**Ablation on Graph Construction and Sharing Strategy (RQ3).** We further ablate the bottleneck construction, i.e., how variable-level states are formed and how the graph is applied (Table 5). *MaxPool* replaces average pooling with max pooling over patches, testing whether emphasizing the strongest local responses is beneficial. *LastPool* uses the last patch token as the window state, testing whether a simple end-state summary suffices. *Per-Patch Graph* removes window-level aggregation and learns a separate graph for each patch position, testing whether patch-specific dependency estimation improves modeling. All variants underperform the default **AvgPool** with a shared graph, in-

dicating that stable window-level aggregation and graph sharing prevent instantaneous perturbations from dominating dependency estimation, thereby yielding more robust cross-variable interaction and better forecasting accuracy.

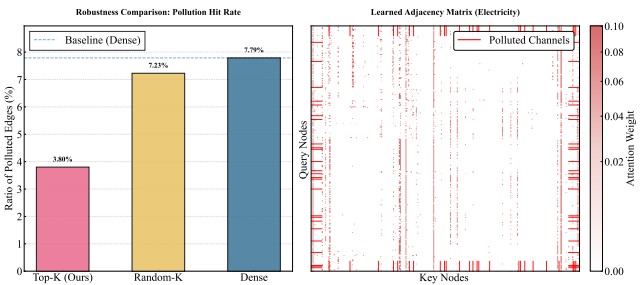

*Figure 3.* Robustness analysis under noisy channel injection. See Appendix G for more examples.

### 4.5. Model Analysis

**Noise-Aware Dependency Selection Analysis.** To evaluate whether MS-FLOW truly selects critical dependencies while ignoring irrelevant variables, we conduct a robustness experiment by injecting artificial noisy channels into the input. Specifically, on the Electricity dataset, we randomly choose 25 channels and add Gaussian noise with intensity 5. We measure the pollution hit rate, defined as the proportion of selected dependency edges that involve noisy channels. A lower value indicates stronger robustness and better dependency discrimination ability. As shown in Fig. 3, MS-FLOW achieves a pollution ratio of only 3.80%, which is substantially lower than Random-K (7.23%) and Dense (7.79%). Notably, although the Dense model considers all possible variable interactions, it suffers from the highest contamination rate, indicating that noise is propagated in an unconstrained manner. Random-K partially alleviates this issue but still lacks targeted selection. In contrast, MS-FLOW consistently avoids noisy variables and concentrates information flow on a small set of reliable dependencies. The learned adjacency matrix further confirms that noisy channels are largely excluded from the sparse dependency graph. These results demonstrate that MS-FLOW does not merely reduce connections, but actively learns to ignore irrelevant variables, validating the effectiveness of the proposed sparse-flow bottleneck.

**Efficiency Analysis.** We evaluate the efficiency of MS-FLOW from two aspects, training time and memory footprint, and compare it on two datasets with significantly different scales: ETTm2 and Traffic. All methods are evaluated under the same batch setting, and the results are shown in Fig. 4. On the small-scale ETTm2 dataset, MS-FLOW achieves both the lowest memory consumption and state-of-the-art forecasting accuracy. On the high-dimensional Traffic dataset, MS-FLOW uses less memory than the strong cross-variable baseline TimeFilter, indicating better scal-

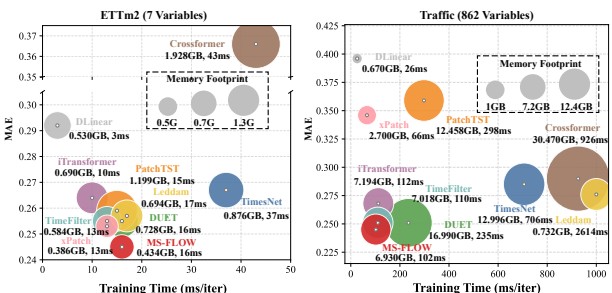

*Figure 4.* Model effectiveness and efficiency comparison on the ETTm2 and Traffic datasets.

ability as the number of variables grows. Notably, MS-FLOW may incur slightly higher resource cost than some lightweight models that discard cross-variable modeling (e.g., xPatch and DLinear) on Traffic, but it delivers substantially better accuracy, demonstrating a favorable trade-off between efficiency and effective cross-variable dependency modeling. Overall, these results highlight MS-FLOW's ability to balance predictive performance and computational cost.

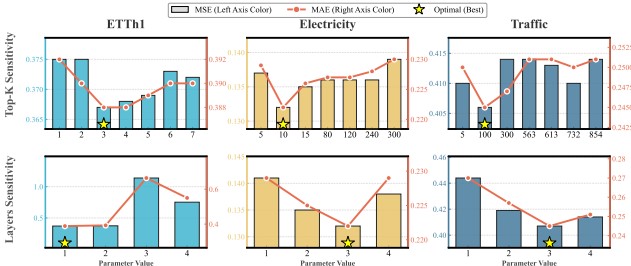

*Figure 5.* Sensitivity analysis of the key hyperparameters in MS-FLOW.

**Hyperparameter Studies.** We analyze the sensitivity of MS-FLOW to two key hyperparameters: the sparsity level $K$ in the sparse bottleneck and the encoder depth $Layers$. As shown in Fig. 5, MS-FLOW achieves its best or near-best performance on all datasets with relatively small $K$ and shallow $Layers$. Specifically, on Electricity, the optimum is reached with only $K = 10$, suggesting that a small subset of variables is sufficient to capture the dominant cross-variable dependencies. On the strongly correlated and high-dimensional Traffic dataset, MS-FLOW attains the best result at $K = 100$, which is still far smaller than the total number of variables, highlighting the effectiveness of sparse routing even in high-dimensional settings. Meanwhile, MS-FLOW favors a shallow architecture, and deeper stacking does not yield consistent gains. Overall, these results indicate that MS-FLOW's advantage does not rely on dense interactions or deep networks; instead, it stems from selectively preserving a few critical dependency paths, supporting the proposed capacity-limited sparse information-flow design.

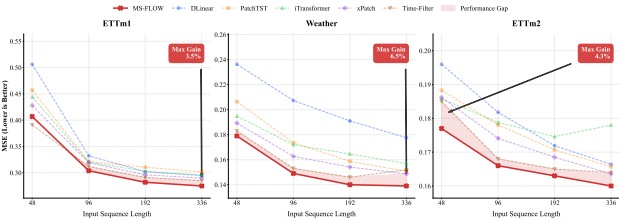

*Figure 6.* Impact of different lookback window lengths on forecasting accuracy across datasets; the shaded regions indicate the maximum performance gap between MS-FLOW and the baseline models.

**Influence of look-back horizon.** We study the influence of lookback window length on forecasting performance to evaluate how effectively MS-FLOW utilizes historical information. Specifically, we progressively increase the input sequence length from 48 to 336, and the results are shown in Fig. 6. Overall, MS-FLOW achieves top-tier performance across different lookback lengths, demonstrating strong robustness to the choice of historical window. Notably, as the input length increases, the forecasting error of MS-FLOW generally decreases, indicating that it can extract useful signals from longer historical contexts. On the Weather dataset, MS-FLOW attains a maximum relative gain of 6.5% over the strongest baseline under this setting. In summary, these results suggest that MS-FLOW can continuously benefit from longer lookback windows by selectively preserving informative key dependencies while suppressing redundant ones, further validating the effectiveness of the proposed sparse-flow bottleneck for modeling long-range temporal contexts.

## 5. Conclusion

We propose MS-FLOW for multivariate time series forecasting. It treats cross-variable modeling as capacity-limited information flow to reduce redundancy, spurious correlations, and over-smoothing. MS-FLOW uses patch-level tokens, mixes time patterns along the patch axis, and injects cross-variable signals through a sparse dependency bottleneck that routes only a few key paths. Experiments show strong results on both long- and short-term forecasting. We believe "Let Forecasting Forget" is a useful view for robust dependency modeling.

## Impact Statement

This paper presents work whose goal is to advance the field of Machine Learning. There are many potential societal consequences of our work, none which we feel must be specifically highlighted here.

## Acknowledgements

This work was supported in part by the following: the National Natural Science Foundation of China under Grant Nos. U24A20219, 62272281, U24A20328, the Yantai Natural Science Foundation under Grant No. 2024JCYJ034, and the Youth Innovation Technology Project of Higher School in Shandong Province under Grant No. 2023KJ212.

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

# A. Dataset Descriptions

We evaluate MS-FLOW on a diverse collection of real-world multivariate time series datasets covering energy systems, traffic networks, and environmental monitoring. These datasets vary significantly in temporal resolution, dimensionality, and dependency structures, providing a comprehensive testbed for cross-variable modeling.

- ❶ **ETT (Electricity Transformer Temperature)**[1](Zhou et al., 2021). The ETT dataset records operational data from electricity transformers deployed in two regions of China over a two-year period (2016–2018). It contains seven variables describing oil temperature and power load conditions. To assess robustness across temporal resolutions, we use both hourly-level datasets (ETTh) and 15-minute-level datasets (ETTm), which exhibit strong periodicity as well as regime-dependent correlations between variables.

- ❷ **Traffic**[2] (Wu et al., 2021).The Traffic dataset measures road occupancy rates collected from hundreds of loop detectors installed on the freeways of the San Francisco Bay Area. The data is sampled at an hourly frequency and spans from 2015 to 2016. Due to network-wide congestion propagation and local disturbances, this dataset presents complex and time-varying inter-sensor dependencies, making it challenging for dense interaction models.

- ❸ **Electricity**[3](Wu et al., 2021). The Electricity dataset contains hourly electricity consumption records for 321 individual clients between 2012 and 2014. Each variable corresponds to a different customer, and correlations across variables are typically sparse and state-dependent, reflecting shared consumption patterns under certain conditions (e.g., peak hours) and weak coupling otherwise.

- ❹ **Weather**[4](Wu et al., 2021). The Weather dataset consists of 21 meteorological variables recorded every 10 minutes throughout 2020 at weather stations in Germany. These variables include temperature, humidity, visibility, and other atmospheric indicators. The dataset is characterized by strong seasonal effects, short-term fluctuations, and heterogeneous cross-variable relationships influenced by evolving weather states.

- ❺ **Solar-Energy**[5](Lai et al., 2018). The Solar-Energy dataset reports solar power generation from 137 photovoltaic plants located in Alabama during 2006, sampled at 10-minute intervals. Power outputs across plants are influenced by shared environmental factors such as sunlight and cloud coverage, while local variations introduce intermittent and noisy dependencies.

- ❻ **PEMS**[6](**Liu et al., 2022**). The PEMS datasets are collected from the Caltrans Performance Measurement System and include four traffic networks (PEMS03, PEMS04, PEMS07, and PEMS08) across different regions in California. Traffic measurements are aggregated every 5 minutes, resulting in 288 time steps per day. These datasets exhibit high dimensionality and strong spatial-temporal dynamics, with correlations that evolve rapidly across traffic conditions.

Detailed information about the datasets is provided in Table 6.

# B. Implementation Details

### B.1. Evaluation metrics

Mean square error (MSE) and Mean absolute error (MAE) are commonly used as measures of forecasting performance in time series forecasting tasks. MSE represents the average of the squared differences between the predicted and actual values, giving more weight to larger deviations. MAE reflects the average of the absolute differences, thus providing a more balanced picture of the overall magnitude of the error. Together, these metrics constitute a comprehensive assessment of

---

[1]https://github.com/zhouhaoyi/ETDataset
[2]https://pems.dot.ca.gov/
[3]https://archive.ics.uci.edu/ml/datasets/ElectricityLoadDiagrams20112014
[4]https://www.bgc-jena.mpg.de/wetter/
[5]https://www.nrel.gov/grid/solar-power-data.html
[6]https://pems.dot.ca.gov/

*Table 6.* Detailed dataset descriptions. Channels denotes the number of channels in each dataset. Dataset Split denotes the total number of time points in (Train, Validation, Test) split respectively. Prediction Length denotes the future time points to be predicted and four prediction settings are included in each dataset. Granularity denotes the sampling interval of time points.

| Tasks | Dataset | Dim. | Series Length | Dataset Size | Frequency | Domain | Periodicity |
|---|---|---|---|---|---|---|---|
| Long-term Forecasting | ETTm1 | 7 | {96, 192, 336, 720} | (34465, 11521, 11521) | 15 min | Temperature | 96 |
| | ETTm2 | 7 | {96, 192, 336, 720} | (34465, 11521, 11521) | 15 min | Temperature | 96 |
| | ETTh1 | 7 | {96, 192, 336, 720} | (8545, 2881, 2881) | 1 hour | Temperature | 24 |
| | ETTh2 | 7 | {96, 192, 336, 720} | (8545, 2881, 2881) | 1 hour | Temperature | 24 |
| | Electricity | 321 | {96, 192, 336, 720} | (18317, 2633, 5261) | 1 hour | Electricity | 24 |
| | Traffic | 862 | {96, 192, 336, 720} | (12185, 1757, 3509) | 1 hour | Transportation | 24 |
| | Weather | 21 | {96, 192, 336, 720} | (36792, 5271, 10540) | 10 min | Weather | 144 |
| Short-term Forecasting | PEMS03 | 358 | {12, 24, 48} | (15617, 5135, 5135) | 5 min | Traffic Network | 96 |
| | PEMS04 | 307 | {12, 24, 48} | (10172, 3375, 3375) | 5 min | Traffic Network | 96 |
| | PEMS07 | 883 | {12, 24, 48} | (16911, 5622, 5622) | 5 min | Traffic Network | 96 |
| | PEMS08 | 170 | {12, 24, 48} | (10690, 3548, 3548) | 5 min | Traffic Network | 96 |

model accuracy. The mathematical definitions are as follows:

$$MSE = \frac{1}{H} \sum_{i=1}^{H} \left( \mathcal{Y}_i - \hat{\mathcal{Y}}_i \right)^2$$
$$MAE = \frac{1}{H} \sum_{i=1}^{H} \left| \mathcal{Y}_i - \hat{\mathcal{Y}}_i \right|$$

(18)

Where $H$ represents the size of the prediction window, $\mathcal{Y}_i$ represents the true value, and $\hat{\mathcal{Y}}_i$ represents the predicted value of the model.

### B.2. Experiment Details

All experiments are conducted on a single NVIDIA RTX 3090 GPU. To ensure reproducibility, we fix the random seed to 2021. Each model is trained for a maximum of 30 epochs, with an early stopping patience of 3 epochs. We optimize all models using the mean squared error (MSE) loss with the Adam optimizer. For MS-FLOW, the number of stacked core blocks is selected from Layers∈{1,2,3,4}. For patching, we set patch-len∈{16,24} and fix the stride to stride=8. In the Patch-wise Temporal Interaction (PTI) module, we use a fixed hidden width that is aligned with the number of patches (i.e., the PTI hidden dimension matches the patch count) to keep the temporal mixing capacity consistent across settings. The batch size is chosen from {16,32,64}. For the sparse dependency bottleneck, the Top-K routing budget is dataset-dependent: we use a relatively larger K for high-dimensional datasets (with more variables) and a smaller K for low-dimensional datasets. Importantly, our sensitivity analysis shows that MS-FLOW is not overly sensitive to large K; competitive performance can be achieved with a modest routing budget, indicating that the benefit comes from selective rather than excessive cross-variable communication. For example, for datasets with fewer than 20 variables (such as ETT), we set $K$ to 5-10; for medium-sized datasets (such as Electricity), we set $K$ to 20-30; and for high-dimensional datasets with stronger channel correlations (such as Traffic, which contains 862 variables), we choose $K = 100$. Following the settings in TFB (Qiu et al., 2024) and TAB (Qiu et al., 2025a), we do not apply the "Drop Last" trick to ensure a fair comparison.

### B.3. Implementation Details of Top-K Bottleneck

**Implementation of the Top-$K$ bottleneck.** Let $\mathcal{S} \in \mathbb{R}^{C \times C}$ denote the similarity logits between variables. We obtain the Top-$K$ indices $\mathcal{I}_c = \text{TopK}(\mathcal{S}_{c,:})$ for each variable $c$ and construct a binary mask $\mathcal{M}_{c,j} = 1$ if $j \in \mathcal{I}_c$, otherwise 0. We then form sparsified logits $\tilde{\mathcal{S}} = \mathcal{S} + (1 - \mathcal{M}) \cdot (-\infty)$ (implemented as a large negative constant, e.g., $-10^9$), and compute the sparse dependency weights by $\mathcal{A} = \text{Softmax}(\tilde{\mathcal{S}})$.

*Table 7.* Performance comparison between MS-FLOW and TimeFilter on benchmark datasets.

| Datasets | MS-FLOW | | TimeFilter | |
|---|---|---|---|---|
| | MSE | MAE | MSE | MAE |
| ETTh1 | **0.415±0.002** | **0.416±0.003** | 0.422±0.004 | 0.428±0.003 |
| ETTh2 | **0.309±0.003** | **0.354±0.004** | 0.366±0.005 | 0.397±0.002 |
| ETTm1 | **0.370±0.004** | **0.375±0.003** | 0.378±0.005 | 0.393±0.004 |
| ETTm2 | **0.264±0.006** | **0.312±0.004** | 0.271±0.005 | 0.323±0.003 |
| Electricity | **0.158±0.002** | **0.246±0.003** | 0.160±0.006 | 0.256±0.006 |
| Solar-Energy | **0.207±0.003** | **0.218±0.003** | 0.223±0.004 | 0.250±0.006 |
| Weather | **0.235±0.004** | **0.258±0.005** | 0.240±0.005 | 0.268±0.006 |
| Traffic | 0.441±0.007 | **0.258±0.005** | **0.409±0.008** | 0.270±0.006 |

**Gradient behavior and stability.** The Top-$K$ indexing is discrete and thus non-differentiable; however, gradients are not required to pass through the index operation. Instead, the model is trained end-to-end through the differentiable Softmax on $\tilde{\mathbf{S}}$: only the retained Top-$K$ logits receive gradients, while masked entries have zero gradients. This hard-sparsification-after-logits strategy is widely used in sparse attention/routing and is stable in practice. We further apply dropout on $\mathbf{A}$ to prevent over-confident edges and improve robustness. At inference, we use the same Top-$K$ masking procedure, ensuring consistency between training and testing.

### B.4. Reversible instance normalization

In order to alleviate the impact of distribution drift, we follow the practice of advanced models (Liu et al., 2024) and introduce a reversible normalization mechanism during preprocessing. The normalization process can be expressed as follows:

$$\mathcal{X}_n = \frac{\mathcal{X} - E[\mathcal{X}]}{\sqrt{Var[\mathcal{X}] + \varepsilon}} \tag{19}$$

Where $E[\mathcal{X}]$ and $Var[\mathcal{X}]$ represent the mean and variance of the original sequence respectively. $\varepsilon$ represents a very small positive number to avoid division by 0.

## C. Full Results

This appendix reports the complete quantitative results for both long-term and short-term forecasting tasks. For long-term forecasting, we evaluate all methods under a fixed input length of $L = 96$ and report the average performance over four forecasting horizons $T \in \{96, 192, 336, 720\}$. For short-term forecasting, all models use an input length of $L = 96$, and results are averaged over forecasting horizons $T \in \{12, 24, 48\}$.

The full results on all datasets are summarized in Tables 8 and 11. The best and second-best results are highlighted in **red** and blue, respectively. Overall, the complete results further confirm the robustness and effectiveness of MS-FLOW across different datasets and forecasting horizons.

To ensure the reliability of the experimental results, all experiments are repeated three times with different random seeds. We conduct the Wilcoxon signed-rank test to evaluate the statistical significance of MS-FLOW compared with the strongest baseline, TimeFilter, across all benchmark datasets. Specifically, we perform paired one-sided tests on the averaged MSE and MAE results. As shown in the statistical summary, MS-FLOW achieves a statistically significant improvement in terms of MAE ($p = 0.0039 < 0.01$). For MSE, the improvement is marginally significant ($p = 0.0664 < 0.1$). It is worth noting that MSE is highly sensitive to outliers; despite this, MS-FLOW outperformed the baseline on 15 out of 16 metrics, validating its consistent effectiveness across various prediction scenarios. Detailed statistics are shown in Tables 7.

*Table 8.* Full results for long-term forecasting. All experiments fix the lookback length $L = 96$. The prediction length set is $H \in \{96, 192, 336, 720\}$. The best result is **red**, the second best result is blue.

| Models | | MS-FLOW (Ours) | | TimeFilter (ICML'2025) | | TimeMosaic (AAAI'2026) | | xPatch (AAAI'2025) | | DUET (KDD'2025) | | Leddam (ICML'2024) | | iTransformer (ICLR'2024) | | Crossformer (ICLR'2023) | | TimesNet (ICLR'2023) | | PatchTST (ICLR'2023) | | DLinear (AAAI'2023) | |
|---|---|---|---|---|---|---|---|---|---|---|---|---|---|---|---|---|---|---|---|---|---|---|---|
| Metric | | MSE | MAE | MSE | MAE | MSE | MAE | MSE | MAE | MSE | MAE | MSE | MAE | MSE | MAE | MSE | MAE | MSE | MAE | MSE | MAE | MSE | MAE |
| ETTh1 | 96 | **0.367** | **0.389** | 0.370 | 0.394 | 0.369 | 0.389 | 0.378 | 0.390 | 0.377 | 0.393 | 0.377 | 0.394 | 0.386 | 0.405 | 0.423 | 0.448 | 0.384 | 0.402 | 0.414 | 0.419 | 0.386 | 0.400 |
| | 192 | **0.408** | **0.401** | 0.413 | 0.420 | 0.416 | 0.415 | 0.434 | 0.421 | 0.429 | 0.425 | 0.424 | 0.422 | 0.441 | 0.436 | 0.471 | 0.474 | 0.436 | 0.429 | 0.460 | 0.445 | 0.437 | 0.432 |
| | 336 | **0.436** | **0.418** | 0.450 | 0.440 | 0.454 | 0.435 | 0.479 | 0.443 | 0.471 | 0.446 | 0.459 | 0.442 | 0.487 | 0.458 | 0.570 | 0.546 | 0.491 | 0.469 | 0.501 | 0.466 | 0.481 | 0.459 |
| | 720 | 0.451 | **0.451** | **0.448** | 0.457 | 0.461 | 0.458 | 0.479 | 0.463 | 0.496 | 0.480 | 0.463 | 0.459 | 0.503 | 0.491 | 0.653 | 0.621 | 0.521 | 0.500 | 0.500 | 0.488 | 0.519 | 0.516 |
| | avg | **0.416** | **0.415** | 0.420 | 0.428 | 0.425 | 0.424 | 0.442 | 0.429 | 0.443 | 0.436 | 0.431 | 0.429 | 0.454 | 0.448 | 0.529 | 0.522 | 0.458 | 0.450 | 0.469 | 0.454 | 0.456 | 0.452 |
| ETTh2 | 96 | **0.228** | **0.299** | 0.283 | 0.337 | 0.286 | 0.332 | 0.288 | 0.334 | 0.296 | 0.345 | 0.292 | 0.343 | 0.297 | 0.349 | 0.745 | 0.584 | 0.340 | 0.374 | 0.302 | 0.348 | 0.333 | 0.387 |
| | 192 | **0.279** | **0.331** | 0.362 | 0.392 | 0.362 | 0.380 | 0.363 | 0.383 | 0.368 | 0.389 | 0.367 | 0.389 | 0.380 | 0.400 | 0.877 | 0.656 | 0.402 | 0.414 | 0.388 | 0.400 | 0.477 | 0.476 |
| | 336 | **0.331** | **0.372** | 0.404 | 0.424 | 0.409 | 0.416 | 0.414 | 0.420 | 0.411 | 0.422 | 0.412 | 0.424 | 0.428 | 0.432 | 1.043 | 0.731 | 0.452 | 0.452 | 0.426 | 0.433 | 0.594 | 0.541 |
| | 720 | **0.386** | **0.419** | 0.407 | 0.433 | 0.396 | 0.422 | 0.428 | 0.442 | 0.412 | 0.434 | 0.419 | 0.438 | 0.427 | 0.445 | 1.104 | 0.763 | 0.462 | 0.468 | 0.431 | 0.446 | 0.831 | 0.657 |
| | avg | **0.306** | **0.355** | 0.364 | 0.396 | 0.363 | 0.387 | 0.373 | 0.395 | 0.372 | 0.397 | 0.372 | 0.398 | 0.383 | 0.406 | 0.942 | 0.684 | 0.414 | 0.427 | 0.387 | 0.407 | 0.559 | 0.515 |
| ETTm1 | 96 | **0.302** | **0.330** | 0.313 | 0.354 | 0.308 | 0.338 | 0.316 | 0.420 | 0.324 | 0.354 | 0.319 | 0.359 | 0.334 | 0.368 | 0.404 | 0.426 | 0.338 | 0.375 | 0.329 | 0.367 | 0.345 | 0.372 |
| | 192 | **0.345** | **0.360** | 0.356 | 0.380 | 0.364 | 0.368 | 0.367 | 0.369 | 0.369 | 0.379 | 0.369 | 0.383 | 0.377 | 0.391 | 0.450 | 0.451 | 0.374 | 0.387 | 0.367 | 0.385 | 0.380 | 0.389 |
| | 336 | **0.376** | **0.382** | 0.386 | 0.402 | 0.394 | 0.390 | 0.395 | 0.391 | 0.404 | 0.402 | 0.394 | 0.402 | 0.426 | 0.420 | 0.532 | 0.515 | 0.410 | 0.411 | 0.399 | 0.410 | 0.413 | 0.413 |
| | 720 | **0.448** | **0.427** | 0.452 | 0.437 | 0.460 | 0.428 | 0.464 | 0.431 | 0.463 | 0.437 | 0.460 | 0.442 | 0.491 | 0.459 | 0.666 | 0.589 | 0.478 | 0.450 | 0.454 | 0.439 | 0.474 | 0.453 |
| | avg | **0.368** | **0.375** | 0.377 | 0.393 | 0.381 | 0.381 | 0.386 | 0.403 | 0.390 | 0.393 | 0.385 | 0.396 | 0.407 | 0.410 | 0.513 | 0.495 | 0.400 | 0.406 | 0.387 | 0.400 | 0.403 | 0.407 |
| ETTm2 | 96 | **0.166** | **0.245** | 0.169 | 0.255 | 0.170 | 0.248 | 0.174 | 0.253 | 0.174 | 0.255 | 0.176 | 0.257 | 0.180 | 0.264 | 0.287 | 0.366 | 0.187 | 0.267 | 0.175 | 0.259 | 0.193 | 0.292 |
| | 192 | **0.231** | **0.291** | 0.235 | 0.299 | 0.236 | 0.291 | 0.241 | 0.297 | 0.243 | 0.302 | 0.243 | 0.303 | 0.250 | 0.309 | 0.414 | 0.492 | 0.249 | 0.309 | 0.241 | 0.302 | 0.284 | 0.362 |
| | 336 | **0.288** | **0.327** | 0.293 | 0.336 | 0.295 | 0.331 | 0.300 | 0.336 | 0.304 | 0.341 | 0.303 | 0.341 | 0.311 | 0.348 | 0.597 | 0.542 | 0.321 | 0.351 | 0.305 | 0.343 | 0.369 | 0.427 |
| | 720 | **0.372** | **0.380** | 0.390 | 0.393 | 0.392 | 0.386 | 0.401 | 0.393 | 0.399 | 0.397 | 0.400 | 0.398 | 0.412 | 0.407 | 1.730 | 1.042 | 0.408 | 0.403 | 0.402 | 0.400 | 0.554 | 0.522 |
| | avg | **0.264** | **0.311** | 0.272 | 0.321 | 0.273 | 0.314 | 0.279 | 0.320 | 0.280 | 0.324 | 0.280 | 0.325 | 0.288 | 0.332 | 0.757 | 0.610 | 0.291 | 0.332 | 0.281 | 0.326 | 0.350 | 0.401 |
| Electricity | 96 | **0.132** | **0.222** | 0.133 | 0.230 | 0.160 | 0.258 | 0.162 | 0.246 | 0.145 | 0.233 | 0.141 | 0.235 | 0.148 | 0.240 | 0.219 | 0.314 | 0.168 | 0.272 | 0.181 | 0.270 | 0.197 | 0.282 |
| | 192 | **0.150** | **0.239** | 0.154 | 0.248 | 0.174 | 0.269 | 0.170 | 0.253 | 0.163 | 0.248 | 0.159 | 0.252 | 0.162 | 0.253 | 0.231 | 0.322 | 0.184 | 0.289 | 0.188 | 0.274 | 0.196 | 0.285 |
| | 336 | **0.162** | **0.251** | 0.162 | 0.261 | 0.189 | 0.283 | 0.185 | 0.268 | 0.175 | 0.262 | 0.173 | 0.268 | 0.178 | 0.269 | 0.246 | 0.337 | 0.198 | 0.300 | 0.204 | 0.293 | 0.209 | 0.301 |
| | 720 | **0.183** | **0.273** | 0.184 | 0.284 | 0.225 | 0.304 | 0.224 | 0.302 | 0.204 | 0.291 | 0.201 | 0.295 | 0.225 | 0.317 | 0.280 | 0.363 | 0.220 | 0.320 | 0.246 | 0.324 | 0.245 | 0.333 |
| | avg | **0.157** | **0.246** | 0.158 | 0.256 | 0.187 | 0.278 | 0.185 | 0.267 | 0.172 | 0.258 | 0.168 | 0.262 | 0.178 | 0.270 | 0.244 | 0.334 | 0.192 | 0.295 | 0.205 | 0.290 | 0.212 | 0.300 |
| Weather | 96 | **0.150** | **0.187** | 0.153 | 0.199 | 0.165 | 0.198 | 0.166 | 0.202 | 0.163 | 0.202 | 0.156 | 0.202 | 0.174 | 0.214 | 0.158 | 0.230 | 0.172 | 0.220 | 0.177 | 0.218 | 0.196 | 0.255 |
| | 192 | **0.198** | **0.233** | 0.202 | 0.246 | 0.215 | 0.243 | 0.211 | 0.243 | 0.218 | 0.252 | 0.207 | 0.250 | 0.221 | 0.254 | 0.206 | 0.277 | 0.219 | 0.261 | 0.225 | 0.259 | 0.237 | 0.296 |
| | 336 | **0.253** | **0.276** | 0.260 | 0.289 | 0.272 | 0.286 | 0.267 | 0.284 | 0.274 | 0.294 | 0.262 | 0.291 | 0.278 | 0.296 | 0.272 | 0.335 | 0.280 | 0.306 | 0.278 | 0.297 | 0.283 | 0.335 |
| | 720 | **0.334** | **0.332** | 0.342 | 0.341 | 0.353 | 0.340 | 0.345 | 0.336 | 0.349 | 0.343 | 0.343 | 0.343 | 0.358 | 0.347 | 0.398 | 0.418 | 0.365 | 0.359 | 0.354 | 0.348 | 0.345 | 0.381 |
| | avg | **0.234** | **0.257** | 0.239 | 0.269 | 0.251 | 0.267 | 0.247 | 0.266 | 0.251 | 0.273 | 0.242 | 0.272 | 0.258 | 0.278 | 0.258 | 0.315 | 0.259 | 0.286 | 0.258 | 0.280 | 0.265 | 0.317 |
| Solar-Energy | 96 | **0.173** | **0.197** | 0.193 | 0.223 | 0.209 | 0.248 | 0.203 | 0.218 | 0.200 | 0.207 | 0.197 | 0.241 | 0.203 | 0.238 | 0.310 | 0.331 | 0.250 | 0.292 | 0.234 | 0.286 | 0.290 | 0.378 |
| | 192 | **0.207** | **0.214** | 0.226 | 0.249 | 0.238 | 0.272 | 0.242 | 0.240 | 0.228 | 0.233 | 0.231 | 0.264 | 0.233 | 0.261 | 0.734 | 0.725 | 0.296 | 0.318 | 0.267 | 0.310 | 0.320 | 0.398 |
| | 336 | **0.221** | **0.227** | 0.235 | 0.261 | 0.255 | 0.277 | 0.258 | 0.248 | 0.262 | 0.244 | 0.241 | 0.268 | 0.248 | 0.273 | 0.750 | 0.735 | 0.319 | 0.330 | 0.290 | 0.315 | 0.353 | 0.415 |
| | 720 | **0.236** | **0.236** | 0.239 | 0.268 | 0.257 | 0.284 | 0.273 | 0.257 | 0.258 | 0.249 | 0.250 | 0.281 | 0.249 | 0.276 | 0.769 | 0.765 | 0.338 | 0.337 | 0.289 | 0.317 | 0.356 | 0.413 |
| | avg | **0.209** | **0.218** | 0.223 | 0.250 | 0.240 | 0.270 | 0.244 | 0.241 | 0.237 | 0.233 | 0.230 | 0.264 | 0.233 | 0.262 | 0.641 | 0.639 | 0.301 | 0.319 | 0.270 | 0.307 | 0.330 | 0.401 |
| Traffic | 96 | 0.407 | **0.245** | **0.375** | 0.251 | 0.424 | 0.269 | 0.521 | 0.346 | 0.407 | 0.251 | 0.426 | 0.276 | 0.395 | 0.268 | 0.522 | 0.290 | 0.462 | 0.285 | 0.544 | 0.359 | 0.650 | 0.396 |
| | 192 | 0.429 | **0.247** | **0.395** | 0.262 | 0.448 | 0.279 | 0.506 | 0.330 | 0.431 | 0.262 | 0.458 | 0.289 | 0.417 | 0.276 | 0.530 | 0.293 | 0.473 | 0.296 | 0.540 | 0.354 | 0.598 | 0.370 |
| | 336 | 0.443 | **0.260** | **0.414** | 0.271 | 0.464 | 0.284 | 0.511 | 0.327 | 0.456 | 0.271 | 0.486 | 0.297 | 0.433 | 0.283 | 0.558 | 0.305 | 0.498 | 0.296 | 0.551 | 0.358 | 0.605 | 0.373 |
| | 720 | 0.481 | **0.278** | **0.445** | 0.289 | 0.495 | 0.301 | 0.540 | 0.338 | 0.509 | 0.289 | 0.498 | 0.313 | 0.467 | 0.302 | 0.589 | 0.328 | 0.506 | 0.313 | 0.586 | 0.375 | 0.645 | 0.394 |
| | avg | 0.440 | **0.258** | **0.407** | 0.268 | 0.458 | 0.283 | 0.520 | 0.335 | 0.451 | 0.268 | 0.467 | 0.294 | 0.428 | 0.282 | 0.550 | 0.304 | 0.485 | 0.298 | 0.555 | 0.362 | 0.624 | 0.383 |

## C.1. Additional Comparisons with Recent Baselines

To further validate the effectiveness of MS-FLOW, we additionally compare it with several recent baseline methods, including TimeEmb, TimeKAN, SOFTS, and TimeXer, on standard long-term forecasting benchmarks. As shown in Table 9, MS-FLOW achieves the best overall performance on most datasets. In particular, it consistently outperforms the compared methods on ETTh2, Electricity, Weather, and Solar-Energy, and remains highly competitive on ETTm1 and ETTm2. These results demonstrate that the proposed sparse dependency bottleneck is effective not only against classical baselines, but also against recent methods with stronger representation learning and cross-variable modeling capabilities. Notably, the improvements are more evident on datasets such as Electricity and Weather, where cross-variable dependencies are more complex, further supporting our motivation that selectively preserving critical interactions is more beneficial than retaining all dense interactions. To further position MS-FLOW with respect to graph-based forecasting methods, we additionally compare it with MAGNN, a representative multiscale graph neural forecasting model. As shown in Table 10, MS-FLOW consistently outperforms MAGNN on all six datasets at the prediction horizon of 96. The gains are especially clear on Weather, Solar, and Electricity, suggesting that the proposed sparse dependency bottleneck provides a more effective and robust mechanism for cross-variable interaction modeling than multiscale graph construction in these settings.

*Table 9.* Additional comparison with recent baseline methods on long-term forecasting benchmarks. All experiments fix the lookback length to $L = 96$. Lower values of MSE and MAE indicate better forecasting performance. The best result is shown in red, and the second best result is shown in blue.

| Models | MS-FLOW (Ours) | | TimeEmb (NIPS'2025) | | TimeKAN (ICLR'2025) | | TimeXer (NIPS'2024) | | SOFTS (NIPS'2024) | |
|---|---|---|---|---|---|---|---|---|---|---|
| Metric | MSE | MAE | MSE | MAE | MSE | MAE | MSE | MAE | MSE | MAE |
| **ETTh1** 96 | 0.367 | 0.389 | 0.366 | 0.387 | 0.367 | 0.395 | 0.377 | 0.397 | 0.381 | 0.399 |
| 192 | 0.408 | 0.401 | 0.417 | 0.416 | 0.414 | 0.420 | 0.425 | 0.426 | 0.435 | 0.431 |
| 336 | 0.436 | 0.418 | 0.457 | 0.436 | 0.445 | 0.434 | 0.457 | 0.441 | 0.480 | 0.452 |
| 720 | 0.451 | 0.451 | 0.459 | 0.460 | 0.444 | 0.459 | 0.464 | 0.463 | 0.499 | 0.488 |
| avg | 0.417 | 0.415 | 0.425 | 0.425 | 0.418 | 0.427 | 0.431 | 0.432 | 0.449 | 0.442 |
| **ETTh2** 96 | 0.228 | 0.299 | 0.277 | 0.328 | 0.290 | 0.340 | 0.289 | 0.340 | 0.297 | 0.347 |
| 192 | 0.279 | 0.331 | 0.356 | 0.379 | 0.375 | 0.392 | 0.370 | 0.391 | 0.373 | 0.394 |
| 336 | 0.331 | 0.372 | 0.400 | 0.417 | 0.423 | 0.435 | 0.422 | 0.434 | 0.410 | 0.426 |
| 720 | 0.386 | 0.419 | 0.416 | 0.437 | 0.443 | 0.449 | 0.429 | 0.445 | 0.411 | 0.433 |
| avg | 0.306 | 0.355 | 0.362 | 0.390 | 0.383 | 0.404 | 0.378 | 0.402 | 0.373 | 0.400 |
| **ETTm1** 96 | 0.302 | 0.330 | 0.304 | 0.343 | 0.322 | 0.361 | 0.309 | 0.352 | 0.325 | 0.361 |
| 192 | 0.345 | 0.360 | 0.354 | 0.373 | 0.357 | 0.383 | 0.355 | 0.378 | 0.375 | 0.389 |
| 336 | 0.376 | 0.382 | 0.379 | 0.393 | 0.382 | 0.401 | 0.387 | 0.399 | 0.405 | 0.412 |
| 720 | 0.448 | 0.427 | 0.435 | 0.428 | 0.445 | 0.435 | 0.448 | 0.435 | 0.466 | 0.447 |
| avg | 0.368 | 0.375 | 0.368 | 0.384 | 0.376 | 0.395 | 0.375 | 0.391 | 0.393 | 0.403 |
| **ETTm2** 96 | 0.166 | 0.245 | 0.163 | 0.242 | 0.174 | 0.255 | 0.171 | 0.255 | 0.180 | 0.261 |
| 192 | 0.231 | 0.291 | 0.226 | 0.285 | 0.239 | 0.299 | 0.238 | 0.300 | 0.246 | 0.306 |
| 336 | 0.288 | 0.327 | 0.286 | 0.324 | 0.301 | 0.340 | 0.301 | 0.340 | 0.319 | 0.352 |
| 720 | 0.372 | 0.380 | 0.383 | 0.381 | 0.395 | 0.396 | 0.401 | 0.397 | 0.405 | 0.401 |
| avg | 0.264 | 0.311 | 0.264 | 0.308 | 0.277 | 0.322 | 0.278 | 0.323 | 0.287 | 0.330 |
| **Electricity** 96 | 0.132 | 0.222 | 0.136 | 0.231 | 0.174 | 0.266 | 0.151 | 0.247 | 0.143 | 0.233 |
| 192 | 0.150 | 0.239 | 0.153 | 0.246 | 0.182 | 0.273 | 0.165 | 0.261 | 0.158 | 0.248 |
| 336 | 0.162 | 0.251 | 0.170 | 0.264 | 0.197 | 0.286 | 0.183 | 0.280 | 0.178 | 0.264 |
| 720 | 0.183 | 0.273 | 0.208 | 0.297 | 0.236 | 0.320 | 0.220 | 0.309 | 0.218 | 0.305 |
| avg | 0.157 | 0.246 | 0.167 | 0.260 | 0.197 | 0.286 | 0.180 | 0.274 | 0.174 | 0.264 |
| **Weather** 96 | 0.150 | 0.187 | 0.150 | 0.190 | 0.162 | 0.208 | 0.168 | 0.209 | 0.166 | 0.208 |
| 192 | 0.198 | 0.233 | 0.200 | 0.238 | 0.207 | 0.249 | 0.220 | 0.254 | 0.217 | 0.253 |
| 336 | 0.253 | 0.276 | 0.259 | 0.282 | 0.263 | 0.290 | 0.276 | 0.296 | 0.282 | 0.300 |
| 720 | 0.334 | 0.332 | 0.339 | 0.336 | 0.338 | 0.340 | 0.353 | 0.347 | 0.356 | 0.351 |
| avg | 0.234 | 0.257 | 0.237 | 0.262 | 0.242 | 0.272 | 0.254 | 0.276 | 0.255 | 0.278 |
| **Solar-Energy** 96 | 0.173 | 0.197 | - | - | - | - | - | - | 0.200 | 0.230 |
| 192 | 0.207 | 0.214 | - | - | - | - | - | - | 0.229 | 0.253 |
| 336 | 0.221 | 0.227 | - | - | - | - | - | - | 0.243 | 0.269 |
| 720 | 0.236 | 0.236 | - | - | - | - | - | - | 0.245 | 0.272 |
| avg | 0.209 | 0.218 | - | - | - | - | - | - | 0.229 | 0.256 |
| **Traffic** 96 | 0.407 | 0.245 | 0.432 | 0.279 | - | - | 0.416 | 0.280 | 0.376 | 0.251 |
| 192 | 0.429 | 0.247 | 0.442 | 0.289 | - | - | 0.435 | 0.288 | 0.398 | 0.261 |
| 336 | 0.443 | 0.260 | 0.456 | 0.295 | - | - | 0.451 | 0.296 | 0.415 | 0.269 |
| 720 | 0.481 | 0.278 | 0.487 | 0.311 | - | - | 0.484 | 0.314 | 0.447 | 0.287 |
| avg | 0.440 | 0.258 | 0.454 | 0.293 | - | - | 0.447 | 0.295 | 0.409 | 0.267 |

# D. Why Variable-level State Compression is Necessary

This appendix provides further justification for the variable-level state compression used in **Bottleneck-1**.

**Motivation.** The goal of cross-variable dependency modeling is to capture stable couplings induced by the current input window, rather than transient correlations caused by local fluctuations at individual patches. However, patch-level tokens are inherently noisy and heterogeneous: different patch positions may correspond to different local regimes, seasonal phases, or short-term disturbances. Directly constructing dependency structures from patch-level representations therefore exposes the model to high-variance and locally biased signals.

**Noise Reduction Perspective.** Let $\mathcal{T}_{c,n} \in \mathbb{R}^D$ denote the token of variable $c$ at patch position $n$. We can decompose it as

$$\mathcal{T}_{c,n} = \bar{\mathcal{T}}_c + \epsilon_{c,n}, \tag{20}$$

*Table 10.* Comparison with the GNN-based method MAGNN at prediction horizon $H = 96$. Lower is better.

| Dataset | MS-FLOW | MAGNN |
|---|---|---|
| ETTh1 | **0.367** | 0.382 |
| ETTm1 | **0.302** | 0.321 |
| Weather | **0.150** | 0.165 |
| Solar | **0.173** | 0.200 |
| Electricity | **0.132** | 0.153 |
| Traffic | **0.407** | 0.431 |

*Table 11.* Full results for short-term forecasting. The input sequence length of all baseline models is set to 96. All results are averaged over four different forecasting horizons: H $\in \{12, 24, 48\}$. The best result is **red**, the second best result is blue.

| Models | | MS-FLOW (Ours) | | TimeFilter (ICML'2025) | | TimeMosaic (AAAI'2026) | | xPatch (AAAI'2025) | | DUET (KDD'2025) | | Leddam (ICML'2024) | | iTransformer (ICLR'2024) | | Crossformer (ICLR'2023) | | TimesNet (ICLR'2023) | | PatchTST (ICLR'2023) | | DLinear (AAAI'2023) | |
|---|---|---|---|---|---|---|---|---|---|---|---|---|---|---|---|---|---|---|---|---|---|---|---|
| Metric | | MSE | MAE | MSE | MAE | MSE | MAE | MSE | MAE | MSE | MAE | MSE | MAE | MSE | MAE | MSE | MAE | MSE | MAE | MSE | MAE | MSE | MAE |
| PEM03 | 12 | **0.060** | **0.161** | 0.063 | 0.165 | 0.069 | 0.169 | 0.085 | 0.195 | 0.064 | 0.166 | 0.068 | 0.174 | 0.071 | 0.174 | 0.090 | 0.203 | 0.085 | 0.192 | 0.099 | 0.216 | 0.122 | 0.243 |
| | 24 | **0.072** | **0.175** | 0.079 | 0.185 | 0.084 | 0.192 | 0.118 | 0.224 | 0.081 | 0.186 | 0.094 | 0.202 | 0.093 | 0.201 | 0.121 | 0.240 | 0.118 | 0.223 | 0.142 | 0.259 | 0.201 | 0.317 |
| | 48 | **0.104** | **0.213** | 0.110 | 0.222 | 0.118 | 0.228 | 0.203 | 0.295 | 0.114 | 0.222 | 0.140 | 0.254 | 0.125 | 0.236 | 0.202 | 0.317 | 0.155 | 0.260 | 0.211 | 0.319 | 0.333 | 0.425 |
| | avg | **0.079** | **0.183** | 0.084 | 0.191 | 0.090 | 0.196 | 0.135 | 0.238 | 0.086 | 0.191 | 0.101 | 0.210 | 0.096 | 0.204 | 0.138 | 0.253 | 0.119 | 0.225 | 0.151 | 0.265 | 0.219 | 0.328 |
| PEM04 | 12 | **0.068** | **0.167** | 0.068 | 0.167 | 0.081 | 0.187 | 0.103 | 0.213 | 0.079 | 0.181 | 0.076 | 0.182 | 0.078 | 0.183 | 0.098 | 0.218 | 0.087 | 0.195 | 0.105 | 0.224 | 0.148 | 0.272 |
| | 24 | **0.077** | **0.178** | 0.080 | 0.183 | 0.099 | 0.208 | 0.114 | 0.248 | 0.096 | 0.203 | 0.097 | 0.209 | 0.095 | 0.205 | 0.131 | 0.256 | 0.103 | 0.215 | 0.153 | 0.257 | 0.224 | 0.340 |
| | 48 | **0.099** | **0.207** | 0.101 | 0.209 | 0.121 | 0.221 | 0.241 | 0.323 | 0.114 | 0.226 | 0.132 | 0.249 | 0.120 | 0.233 | 0.205 | 0.326 | 0.136 | 0.250 | 0.229 | 0.339 | 0.335 | 0.437 |
| | avg | **0.081** | **0.184** | 0.083 | 0.186 | 0.100 | 0.205 | 0.153 | 0.261 | 0.096 | 0.203 | 0.102 | 0.213 | 0.098 | 0.207 | 0.145 | 0.267 | 0.109 | 0.220 | 0.162 | 0.273 | 0.236 | 0.350 |
| PEM07 | 12 | **0.053** | **0.142** | 0.055 | 0.150 | 0.072 | 0.175 | 0.078 | 0.187 | 0.060 | 0.156 | 0.066 | 0.164 | 0.067 | 0.165 | 0.094 | 0.200 | 0.082 | 0.181 | 0.095 | 0.207 | 0.115 | 0.242 |
| | 24 | **0.068** | **0.160** | 0.068 | 0.166 | 0.085 | 0.185 | 0.116 | 0.215 | 0.073 | 0.172 | 0.079 | 0.185 | 0.088 | 0.190 | 0.139 | 0.247 | 0.101 | 0.204 | 0.150 | 0.262 | 0.210 | 0.329 |
| | 48 | **0.087** | **0.181** | 0.089 | 0.193 | 0.117 | 0.236 | 0.197 | 0.279 | 0.096 | 0.201 | 0.115 | 0.228 | 0.110 | 0.215 | 0.311 | 0.369 | 0.134 | 0.238 | 0.253 | 0.340 | 0.398 | 0.458 |
| | avg | **0.069** | **0.161** | 0.071 | 0.170 | 0.091 | 0.199 | 0.130 | 0.227 | 0.076 | 0.176 | 0.087 | 0.192 | 0.088 | 0.190 | 0.181 | 0.272 | 0.106 | 0.208 | 0.166 | 0.270 | 0.241 | 0.343 |
| PEM08 | 12 | **0.062** | **0.154** | 0.064 | 0.162 | 0.081 | 0.192 | 0.097 | 0.204 | 0.072 | 0.168 | 0.070 | 0.173 | 0.079 | 0.182 | 0.165 | 0.214 | 0.112 | 0.212 | 0.168 | 0.232 | 0.154 | 0.276 |
| | 24 | 0.085 | **0.176** | **0.079** | 0.182 | 0.098 | 0.211 | 0.144 | 0.240 | 0.093 | 0.191 | 0.091 | 0.200 | 0.115 | 0.219 | 0.215 | 0.260 | 0.141 | 0.238 | 0.224 | 0.281 | 0.248 | 0.353 |
| | 48 | 0.117 | **0.203** | **0.105** | 0.214 | 0.151 | 0.282 | 0.237 | 0.304 | 0.123 | 0.217 | 0.145 | 0.261 | 0.186 | 0.235 | 0.315 | 0.335 | 0.198 | 0.283 | 0.321 | 0.354 | 0.440 | 0.470 |
| | avg | 0.088 | **0.178** | **0.083** | 0.186 | 0.110 | 0.228 | 0.159 | 0.249 | 0.096 | 0.192 | 0.102 | 0.211 | 0.127 | 0.212 | 0.232 | 0.270 | 0.150 | 0.244 | 0.238 | 0.289 | 0.281 | 0.366 |

where $\bar{\mathcal{T}}_c$ represents the underlying variable-level state over the input window, and $\epsilon_{c,n}$ captures patch-specific deviations such as local noise or short-lived perturbations. Applying average pooling along the patch dimension yields

$$\mathcal{H}_c = \frac{1}{N}\sum_{n=1}^{N}\mathcal{T}_{c,n} = \bar{\mathcal{T}}_c + \frac{1}{N}\sum_{n=1}^{N}\epsilon_{c,n}. \tag{21}$$

Under mild assumptions that $\epsilon_{c,n}$ has zero mean or weak temporal correlation, the aggregation suppresses patch-level noise and produces a lower-variance estimator of the variable state. This makes subsequent dependency estimation more robust to local disturbances.

**Bias–Variance Trade-off in Dependency Learning.** Constructing cross-variable dependencies at the patch level introduces $N$ times more degrees of freedom than variable-level modeling. While this increases expressive capacity, it also substantially raises the risk of overfitting, especially in high-dimensional settings. By compressing patch tokens into a single variable-level state, MS-FLOW reduces dependency learning to window-level degrees of freedom, achieving a more favorable bias–variance trade-off. This design encourages the model to focus on persistent variable couplings that are predictive across the window, rather than reacting to ephemeral correlations.

**Interpretability and Structural Consistency.** Another benefit of variable-level compression is structural consistency. Since all patch positions of a variable share the same dependency structure, the learned graph reflects global variable interactions within the window. This not only improves interpretability, but also aligns with the objective of the sparse dependency bottleneck, which aims to regulate information flow at the variable level rather than at individual temporal fragments.

In summary, variable-level state compression serves three purposes: (1) it suppresses patch-level noise and short-term

---

**Algorithm 1** MS-FLOW: Sparse Information Flow for Multivariate Forecasting

---

**Require:** Input sequence $\mathcal{X} \in \mathbb{R}^{B \times L \times C}$, patch length $P$, stride $S$, sparsity level $K$
**Ensure:** Forecast $\hat{\mathcal{Y}} \in \mathbb{R}^{B \times H \times C}$
 1: Normalize input using RevIN: $\mathcal{X} \leftarrow \text{RevIN}(\mathcal{X})$
 2: Segment $\mathcal{X}$ into overlapping patches and embed: $\mathcal{Z} \leftarrow \text{PatchEmbed}(\mathcal{X}; P, S)$
 3: $\mathcal{Z} \in \mathbb{R}^{B \times C \times N \times D}$
 4: **for** each encoder layer **do**
 5:     **(Temporal Interaction)**
 6:     Mix patches within each variable using MLP
 7:     **(Sparse Dependency Bottleneck)**
 8:     Aggregate patches to variable-level states
 9:     Learn sparse cross-variable dependencies (Top-$K$)
10:     Propagate information only along selected dependencies
11:     **(Feature Refinement)**
12:     Apply feed-forward transformation
13: **end for**
14: Flatten patch representations and project to future horizon
15: De-normalize output using RevIN
16: **return** $\hat{\mathcal{Y}}$

---

perturbations, (2) it reduces the complexity of dependency learning to mitigate overfitting, and (3) it enforces structurally consistent and interpretable cross-variable interactions. These properties jointly motivate the design choice of **Bottleneck-1** in MS-FLOW.

## E. More details about MS-FLOW

This appendix provides a detailed description of how MS-FLOW is implemented and executed in practice, complementing the high-level formulation in the main paper. Given an input multivariate time series, MS-FLOW first applies instance-wise normalization and segments each variable into overlapping temporal patches, which are embedded into patch-level tokens shared across variables. These patch tokens are then processed by a stack of GraphPatchBlocks.

Within each block, MS-FLOW performs three steps sequentially. First, patch-wise temporal interaction mixes tokens along the patch dimension independently for each variable, stabilizing temporal representations before cross-variable modeling. Second, variable-level state compression aggregates patch tokens of each variable into a compact window-level representation, which is used to estimate a sample-adaptive sparse dependency graph under a capacity constraint. This graph is shared across all patches and governs cross-variable information propagation through sparse routing. Finally, a lightweight feed-forward network refines the features at each patch position. Residual connections and normalization are applied after each stage to ensure stable training.

After passing through multiple blocks, the resulting representations are flattened along the patch and feature dimensions and projected to future horizons via a shared linear prediction head, followed by inverse normalization to recover the original scale. The overall forward process is summarized in Algorithm 1, which presents a concise and implementation-oriented view of MS-FLOW.

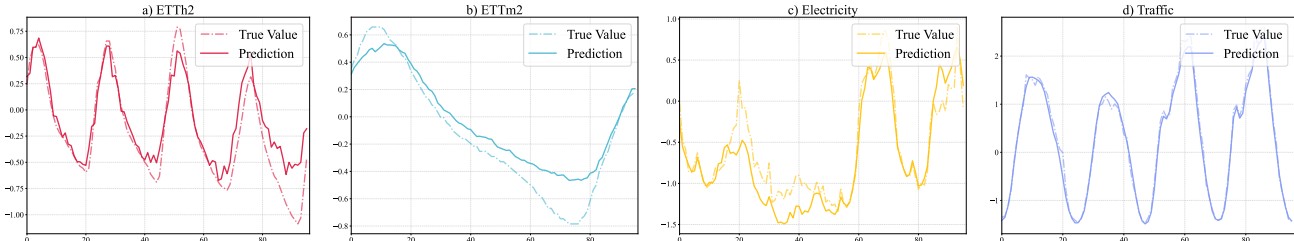

*Figure 7.* We visualize the forecasting results on ETTh2, ETTm2, ECL, and Traffic with a prediction horizon of 96. For each dataset, the ground truth is shown as a solid line and the prediction as a dashed line.

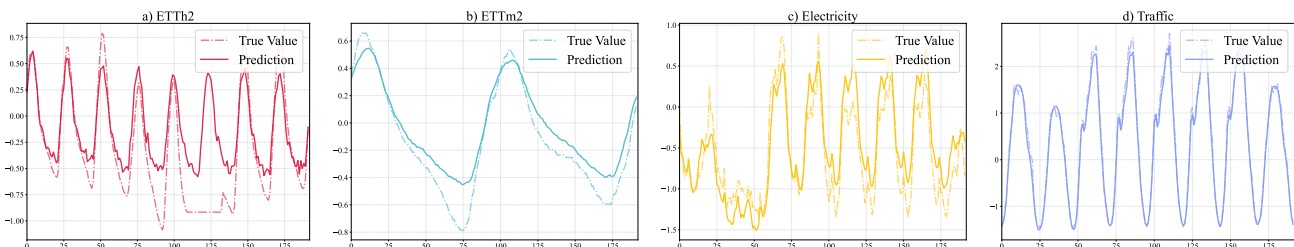

*Figure 8.* We visualize the forecasting results on ETTh2, ETTm2, ECL, and Traffic with a prediction horizon of 192. For each dataset, the ground truth is shown as a solid line and the prediction as a dashed line.

## F. Visualization of prediction results

To qualitatively evaluate the forecasting behavior of MS-FLOW, we visualize the prediction results on four representative datasets, including ETTh2, ETTm2, Electricity (ECL), and Traffic, under multiple forecasting horizons. Fig. 7–10 present the forecasting curves with prediction lengths of 96, 192, 336, and 720, respectively. For each dataset, the ground-truth time series is shown as a solid line, while the corresponding predictions are depicted as dashed lines. As can be observed, MS-FLOW is able to accurately track the overall temporal dynamics across different horizons and datasets. In particular, the model preserves the dominant trends and periodic patterns on ETTh2 and ETTm2, while remaining robust to the irregular fluctuations and noise in Electricity and Traffic. As the forecasting horizon increases, although the prediction uncertainty naturally grows, MS-FLOW still maintains stable alignment with the ground truth without severe drift or error accumulation. These qualitative results complement the quantitative evaluations reported in the main paper, further demonstrating that MS-FLOW effectively captures long-range temporal dependencies while suppressing spurious variations, leading to robust and consistent forecasting performance across both short-term and long-term horizons.

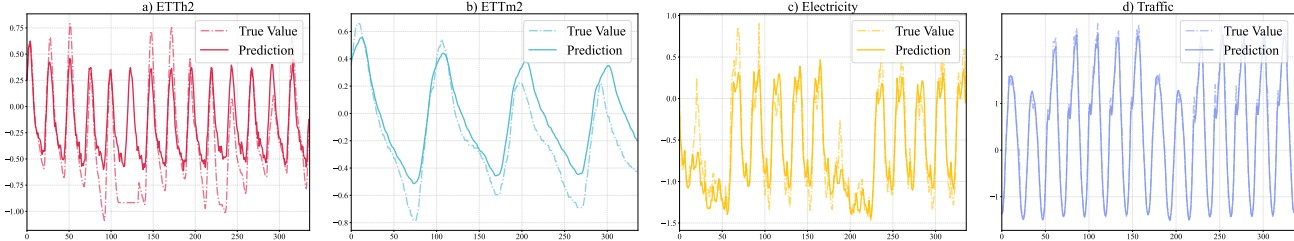

*Figure 9.* We visualize the forecasting results on ETTh2, ETTm2, ECL, and Traffic with a prediction horizon of 336. For each dataset, the ground truth is shown as a solid line and the prediction as a dashed line.

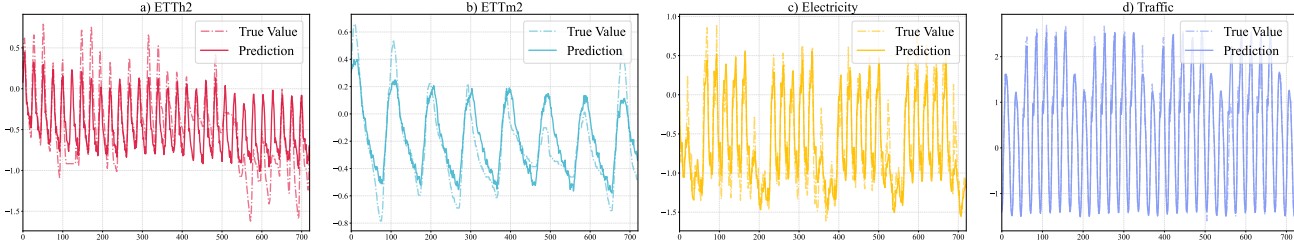

*Figure 10.* We visualize the forecasting results on ETTh2, ETTm2, ECL, and Traffic with a prediction horizon of 720. For each dataset, the ground truth is shown as a solid line and the prediction as a dashed line.

## G. Time series channel correlation analysis

### G.1. Additional Noise-Robustness Analysis

To further examine the noise robustness of MS-FLOW, we extend the noise-injection study to four subsets of the PEM dataset. Following the same protocol as in the main paper, we randomly inject artificial noise into a subset of channels

and measure how often these noisy variables are selected in the learned dependency structure (i.e., the pollution hit rate). Across all PEM subsets, MS-FLOW consistently achieves the lowest contamination rate compared with Random-K and Dense baselines, and the advantage remains clear even under relatively large Top-K settings. Notably, this Top-K is still far smaller than the total number of variables, yet sufficient to capture the major cross-variable dependencies. This indicates that MS-FLOW does not rely on overly aggressive sparsification; instead, it benefits from selective information routing that preserves informative connections while suppressing irrelevant or noise-corrupted ones. The structural visualization provides further evidence. As shown in Fig. 11, the sparse adjacency learned by MS-FLOW contains substantially fewer connections involving polluted channels, whereas Random-K and Dense exhibit much broader noise contamination. Overall, the consistent patterns across all PEM subsets demonstrate that MS-FLOW can robustly identify critical dependencies and effectively ignore irrelevant variables, validating the proposed sparse-flow bottleneck under varying data distributions.

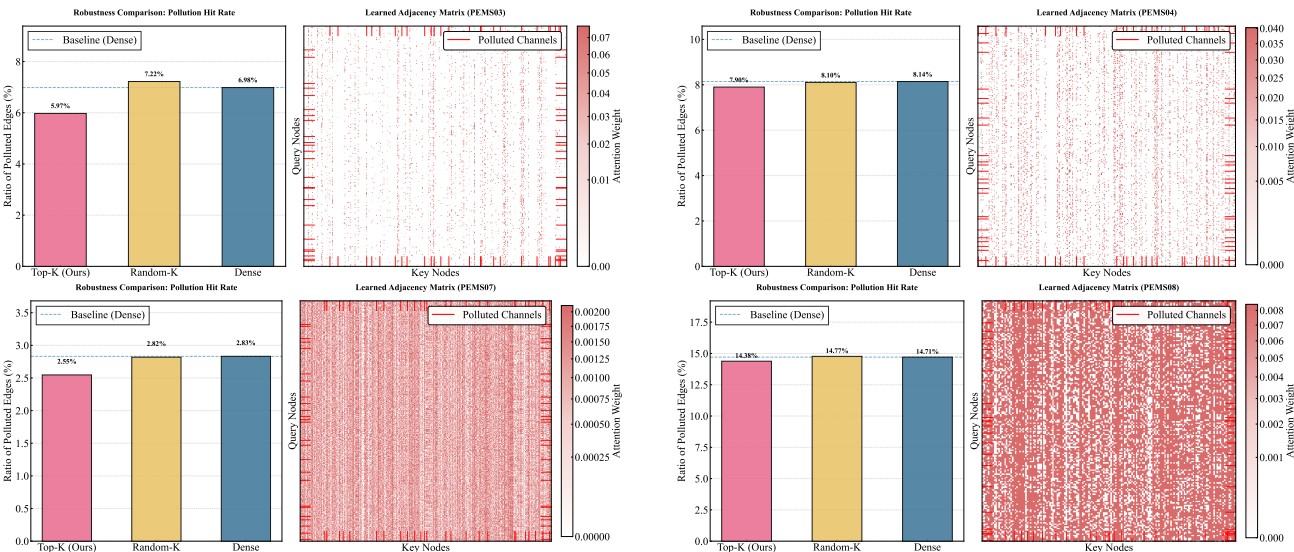

*Figure 11.* Additional noise-robustness analysis on four PEM subsets. For each subset, we compare the pollution hit rate of Top-K , Random-K, and Dense strategies, and visualize the corresponding learned adjacency matrices with polluted channels highlighted (right).

### G.2. Analysis of multivariate correlations

By explicitly modeling cross-variable interaction as sparse information flow, MS-FLOW yields dependency maps with strong interpretability.

We visualize the evolution of multivariate dependency structures on the Weather dataset, where both historical and future windows exhibit clear correlation patterns (Fig. 12). At early encoding stages, the learned dependency maps closely resemble the correlation structure of the input window, indicating that the model initially captures state-dependent couplings present in the observed history.

As the encoding depth increases, the dependency maps are progressively reshaped and begin to align with the correlation structure of the future sequence. This transition suggests that MS-FLOW does not simply preserve input correlations, but selectively filters and reorganizes them under the guidance of the prediction objective.

Importantly, this behavior emerges naturally from the proposed sparse-flow bottleneck: by restricting information propagation to a limited set of critical paths, the model is encouraged to discard transient or noisy correlations and retain dependencies that are most relevant for forecasting. As a result, the learned dependency structure evolves from reflecting past states to approximating future-oriented relationships, providing a mechanistic explanation for the interpretability of MS-FLOW.

### G.3. Limitations

Although MS-FLOW achieves consistent and strong performance across a wide range of benchmarks, several limitations remain. First, our experiments mainly focus on standard multivariate time series datasets whose channel sizes range from tens to a few hundred. Under these settings, a fixed or empirically selected sparsity level $K$ is sufficient to balance modeling

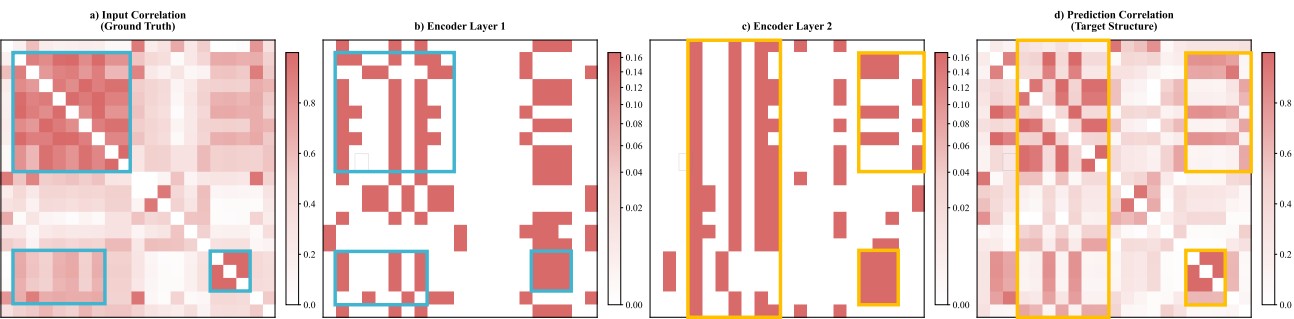

*Figure 12.* Visualization of correlation structures and learned dependency maps. We compare the input correlation, intermediate dependency maps learned by MS-FLOW, and the target correlation. The gradual transition illustrates how cross-variable dependencies are progressively reshaped during encoding.

capacity and efficiency. However, in more extreme high-dimensional scenarios (e.g., datasets with thousands of variables or more), such dataset-level heuristics for choosing $K$ may no longer be optimal. While such ultra-high-dimensional time series are relatively rare in practice, they can arise in certain industrial monitoring systems or large-scale sensor networks.

Second, MS-FLOW currently requires specifying the sparsity budget $K$ at the dataset level, rather than learning it in a fully adaptive manner. Designing a principled mechanism to automatically allocate the routing capacity—based on the number of variables, correlation strength, or the complexity of the current input state—remains an open and important direction. For example, incorporating adaptive $K$ estimation strategies guided by statistical correlations, information-theoretic measures (e.g., entropy), or spectral properties may further improve robustness and generalization in extreme-scale settings.

In summary, developing dynamic and theoretically grounded sparsity control for ultra-high-dimensional multivariate forecasting is a promising future direction to further extend the applicability of MS-FLOW.

