# OpenReview forum: "What If We Let Forecasting Forget? A Sparse Bottleneck for Cross-Variable Dependencies"
_ICML.cc/2026/Conference — ICML 2026 regular_

### Official Review · Reviewer_krsv · 2026-02-20

**Soundness:** 3
**Presentation:** 3
**Significance:** 2
**Originality:** 3
**Overall Recommendation:** 3
**Confidence:** 3

**Summary:**

This paper proposes MS-FLOW, where, instead of using fully connected layers, they do sparse routing to reduce "redundant connections". The model architecture consists of 3 components: Temporal Patch Encoding, Patch-wise Temporal Interaction (a channel-independent MLP over patch tokens), and a Sparse Dependency Bottleneck (Top-K graph estimation + message passing).

**Compliance With Llm Reviewing Policy:**

Affirmed.

**Key Questions For Authors:**

- **Q1**: The paper argues that dense cross-variable interactions propagate spurious correlations, but "spurious" has not been properly defined. How do the authors decide which dependency is spurious or not (e;g;, theoretical justification rather than directly conducting top-K selection)? What is spurious?
- **Q2**: The model performs multivariate forecasting using a shared architecture across all target variables and emphasizes a relatively small model size. I am wondering what the trade-off is for having all the predicted variables share the same model structure. Some variables in the input might hold completely different characteristics e.g., seasonality and noise ratio, and might rely on different inter-variable dependencies. How does the model handle such heterogeneity, and does enforcing a compact shared structure hurt performance for some variables? A per-variable analysis or ablation on partial parameter sharing would help clarify this.
- **Q3**: The sparsity level K seems to be the key component for the predictability, yet it seems to be very task-specific, with the sweet spot different for each dataset. How should one select K for an unseen dataset without grid search?

**Limitations:**

yes

**Strengths And Weaknesses:**

**Strengths**
- **S1**: The presentation of the paper is clear and easy to follow.
- **S2**: The experiment is extensive. The authors provided both qualitative and quantitative analysis.


**Weaknesses**
- **W1**: The method builds upon the assumption that there is over-interaction between variables but there is no theoretical or statistical analysis (e.g., using mutual information, Granger causality, or using controllable synthetic data) on whether such an assumption is the true case. Without such evidence, it remains unclear whether the performance gains are from the sparse bottleneck enforcing the right inductive bias, or simply from implicit regularization introduced by Top-K sparsification.
- **W2**: Many datasets the authors have used do not necessarily contain informative dependencies (e.g., ETTh, Weather, and Traffic) [1], thus using such datasets casts doubt on whether removing dependencies on datasets that contain more nuanced interactions is necessary.

[1] https://cbergmeir.com/talks/neurips2024/

---

> ### Author Rebuttal · Authors · 2026-03-31
>
> We appreciate your valuable suggestions and provide our detailed responses below.
>
> W1: We appreciate the valuable feedback from the reviewers. We agree that a more explicit validation of the “over-interaction” hypothesis would strengthen the paper. To this end, we added a dependency concentration analysis on the Electricity dataset. As shown in Fig. 1 at the anonymous link, the cumulative dependency ratio rises rapidly within the first few neighbors, indicating that most effective dependencies concentrate on a small subset of key variables rather than being uniformly distributed across all variables. This supports our claim that not all interactions are equally important. In addition, Section 5.4 (Noise-Aware Dependency Selection Analysis) already shows that dense interactions introduce more noise-contaminated dependencies, while random or unstructured sparsification performs worse than our Top-K routing strategy. Together, these results suggest that the gains do not merely come from implicit regularization, but from the structured and selective information flow enforced by the sparse bottleneck.
>
> W2: Thank you for the reviewers' comments. We agree that many datasets may not contain extremely rich dependency structures, but this does not conflict with our central argument. On the contrary, it supports our claim that not all interactions need to be explicitly preserved for effective forecasting. As shown in Fig. 1, cross-variable connections often contain redundant, weak, or noisy dependencies, and retaining all of them may lead to harmful information propagation. Our results in Fig. 5 show that a small number of carefully selected interactions is usually sufficient to capture the dominant cross-variable dependencies. Moreover, Figs. 3 and 11 further show that excessive interactions are more likely to introduce noisy or polluted edges, whereas sparse routing concentrates information flow on more reliable paths. This explains why MS-FLOW achieves strong performance even with only a small number of retained interactions.
>
> Q1: We appreciate the comprehensive and rigorous comments provided by the reviewers. We agree that the term “spurious” correlation should be defined more clearly. In this paper, we use it to refer to cross-variable dependencies that may show statistical correlation but do not help prediction, and may even hurt performance due to noise propagation or overfitting. Such relationships often arise from shared trends, coincidental alignment, or indirect effects, rather than stable predictive relationships. MS-FLOW does not explicitly identify them using predefined criteria; instead, it suppresses them implicitly through a capacity-limited selection mechanism. As shown in Fig. 1 at the anonymous link, most effective dependencies concentrate on a few key variables. By restricting each variable to interact with only a few of the most relevant variables, the model preserves dependencies that are consistently useful for prediction while weakening weak or unstable ones. We will clarify this definition in the revision.
>
> Q2: We appreciate the valuable suggestions from the reviewers. We agree that a key challenge in multivariate time series forecasting is balancing the regularization benefits of a shared architecture with the need to model variable heterogeneity. A fully shared compact structure helps prevent overfitting to noisy patterns of individual variables, although it may be less expressive for highly localized variable-specific behaviors. MS-FLOW mitigates this issue in two ways: first, RevIN independently normalizes each variable at the input layer; second, the sparse dependency bottleneck dynamically generates a sample-adaptive routing graph conditioned on the current input, allowing different variables to be assigned different neighbors rather than sharing one fixed interaction pattern. Following the reviewer’s suggestion, we added parameter-sharing ablations (see Table 1 at the anonymous link) with fully shared, partially shared via K-Means grouping, and fully unshared settings. The results show that partially shared and fully unshared variants bring only limited or even negative performance changes, while significantly increasing memory and training cost. This suggests that the current compact shared design achieves a better balance among accuracy, efficiency, and stability.
>
> Q3: Thank you to the reviewer for raising this important question. Please see our response to W3 in Reviewer 1 (wjSP).
>
> Once again, we sincerely thank you for your time and valuable comments. We hope these additional experiments and clarifications address your concerns, and we kindly ask you to reconsider your assessment of our work in the final evaluation.
>
> Anonymous link: https://anonymous.4open.science/r/ICML-4B7E

---

> > ### Author Rebuttal · Reviewer_krsv · 2026-04-02
> >
> > I appreciate the effort the authors have devoted during the rebuttal. The conclusion is consistent with the conclusion from DLinear and PatchTST, where, for some models, autoregressive modelling is good enough. However, the limitation with the currently selected datasets still persists. Beyond the conclusion "not all interactions need to be explicitly preserved, "I believe the model would benefit from training with datasets that present higher inter-variable dependencies. For example, as suggested in the link that I mentioned, ETT is often related to weather. Additionally, datasets with causality between variables would also be a good choice, like datasets in geospatial domains, where, unlike the traffic dataset, the variables are actually inter-correlated.

---

> > > ### Author Response · Authors · 2026-04-03
> > >
> > > We thank the reviewer for this insightful suggestion. We also agree that MS-FLOW is likely to show more pronounced advantages in scenarios with stronger cross-variable dependencies, since its core design is specifically aimed at selective cross-variable modeling. Furthermore, we conducted an analysis of the variable correlations in the datasets themselves. As shown in Fig. 5 (Anonymous link: https://anonymous.4open.science/r/ICML-4B7E ), cross-variable correlations are indeed commonly present, and are especially more evident in datasets such as Traffic and PEMS, where clear cross-variable correlation structures can be observed. This is also consistent with the findings reported in prior works such as TFB [1], TimeFilter [2], TimeXer [3], and iTransformer [4]. In contrast, the ETT datasets contain fewer variables and relatively weaker cross-variable relationships, so the benefits brought by cross-variable modeling are naturally less pronounced than in high-dimensional and strongly correlated scenarios.
> > >
> > > Following your suggestion, we further added an additional validation experiment in the geospatial domain, using an air quality dataset consisting of observations from two monitoring stations over four years. This dataset exhibits more explicit spatial correlations and potential inter-variable dependencies, and thus provides more direct evidence for evaluating the applicability of MS-FLOW in data with a geospatial and potentially causal background. The results are reported in Table 4 (Anonymous link: https://anonymous.4open.science/r/ICML-4B7E). The experimental results show that MS-FLOW achieves stable and competitive performance across different prediction horizons, which further supports our conclusion that, when stronger variable couplings exist, the selective cross-variable modeling mechanism in MS-FLOW can be effectively leveraged.
> > >
> > >
> > > Regarding the potential relationship between ETT and weather, as well as more broadly geospatial datasets with explicit causal relations among variables, we agree that these are highly valuable directions for future study. We will explicitly discuss this point in the revised manuscript as part of the discussion and future work, and consider it an important direction for further validating the applicability of MS-FLOW.
> > >
> > >
> > > [1]	Qiu, X., Hu, J., Zhou, L., Wu, X., Du, J., Zhang, B., Guo,C., Zhou, A., Jensen, C. S., Sheng, Z., and Yang, B. Tfb:Towards comprehensive and fair benchmarking of time series forecasting methods. VLDB 2024.
> > >
> > >
> > > [2]	Hu, Y., Zhang, G., Liu, P., Lan, D., Li, N., Cheng, D., Dai, T., Xia, S.-T., and Pan, S. Timefilter: Patch-specific spatial-temporal graph filtration for time series forecasting. ICML, 2025.
> > >
> > >
> > > [3]	Wang, Y., Wu, H., Dong, J., Qin, G., Zhang, H., Liu, Y., Qiu, Y., Wang, J., and Long, M. Timexer: Empowering transformers for time series forecasting with exogenous variables. NIPS 2024.
> > >
> > >
> > > [4] Liu, Y., Hu, T., Zhang, H., Wu, H., Wang, S., Ma, L., and Long, M. itransformer: Inverted transformers are effective for time series forecasting. ICLR 2024.
> > >
> > > We sincerely thank the reviewer again for the careful reading of our work and for the thoughtful and constructive feedback. We greatly appreciate the time and effort you have devoted to evaluating our paper, and we kindly hope that the clarifications and additional analyses provided in the rebuttal may encourage you to reconsider your current score.

---

### Official Review · Reviewer_Q1ew · 2026-03-05

**Soundness:** 2
**Presentation:** 2
**Significance:** 2
**Originality:** 2
**Overall Recommendation:** 5
**Confidence:** 3

**Summary:**

The paper addresses the prediction of multivariate time series (MTS), datasets in which many variables are recorded simultaneously over time.
Previous models solve this problem in two fundamentally different ways. Either they ignore all relationships between variables entirely and treat each variable in isolation, or they attempt to learn as many relationships between all variables as possible simultaneously. The authors show that both approaches fail: the first overlooks useful information, while the second learns too many spurious relationships.
Many of these relationships are coincidental or only present in the short term. When a model learns such spurious relationships, its predictions deteriorate.
As a solution, the authors propose a new model called MS-FLOW (Multivariate Sparse Flow), which computes, for each new input, which variables are currently behaving similarly. Each variable is then permitted to receive information only from its most similar neighbors, while all other connections are cut off. Since relevant relationships can change over time, the model makes this selection anew for each input.
Across twelve real-world datasets, MS-FLOW achieves first place on 22 out of 24 evaluation metrics compared to ten competing models.

**Compliance With Llm Reviewing Policy:**

Affirmed.

**Final Justification:**

The rebuttal was thorough and addressed most concerns. The Traffic dataset explanation is reasonable, the baseline count is clarified, SOFTS and TimeXer were added, and the over-smoothing claim is now supported by direct representation-level measurements (cosine similarity and Pearson correlation across encoder layers under dense vs. sparse routing). This last point was my main open concern, and the follow-up experiment provides convincing evidence. I raise my score from 3 to 4.

Remaining issues for the camera-ready revision:

-Notation inconsistencies (L used for both input length and encoder depth, d_k unspecified, node_feat vs. H mismatch)
-Missing comparison with spatiotemporal GNN methods (Graph WaveNet, MTGNN)
-Manual tuning of K remains a practical limitation

**Key Questions For Authors:**

- Why does MS-FLOW perform worse than TimeFilter on the Traffic dataset?

- Are there quantitative measurements supporting the claimed over-smoothing under dense mixing?

- Are there approaches for determining K automatically rather than tuning it manually per dataset?

- Were the learned sparse graphs validated against known physical relationships, for example on the Weather dataset?

- Do the Top-K edges stabilize during training, or do they change continuously?

**Limitations:**

yes

**Strengths And Weaknesses:**

Soundness:

The method works, and the broad evaluation across 12 datasets supports this. However, the statistical significance of the improvements is weak, and on the Traffic dataset MS-FLOW performs worse than the strongest baseline, which contradicts the central thesis but is not discussed. The claim that dense mixing leads to over-smoothing is never empirically substantiated. There is also an inconsistency in the text: the introduction refers to 11 baselines, while Section 4.1 lists only 10. It remains unclear whether this reflects an error or whether a baseline was removed.

-------------------------------
Presentation:

The paper is well structured and the figures are helpful. Figure 1 illustrates the core idea clearly, and Figure 2 provides a good overview of the architecture. The tables are well organized and facilitate comparison with the baselines. However, there are several notation issues. The variable L is used both for the input length (Section 3) and for the encoder depth (Section 4.5). The projection dimension d_k is introduced in Equation 8, but its concrete value is never specified.In Figure 2, the term "node_feat" appears (in Q = W_q · node_feat and K = W_k · node_feat), which does not occur in the equations in the main text, where the corresponding variable is denoted H. Additionally, some formulas in Figure 2 are not legible due to incorrect font embedding in the PDF. The related work section in the main text is kept brief, with the detailed discussion deferred to Appendix F. A comparison with spatiotemporal GNN methods that also learn sparse adjacency matrices is missing.

-------------------------------
Significance:

Modeling cross-variable dependencies in multivariate time series is a relevant problem. The perspective that too much interaction can be harmful is a useful prompt for thought that could influence future work on cross-variable modeling. However, the contribution to understanding is only modest, as no deeper theoretical insight is provided into why Top-K in particular helps. On the practical side, MS-FLOW consistently delivers strong results, but the need to tune K manually per dataset limits its direct applicability. The interpretability claim is never validated against ground-truth structures. The scope is restricted to standard benchmarks, and non-standard settings such as irregular sampling or datasets with thousands of variables are not examined.

-------------------------------
Originality:

The individual components (patch embedding from PatchTST, Top-K sparse attention, average pooling) are all well established. No new tasks, theories, or datasets are introduced. Framing cross-variable interaction as capacity-limited information flow is a coherent conceptual contribution, but not fundamentally novel, as sparse graph learning for time series has already been explored in GNN methods such as Graph WaveNet and MTGNN. Neither of these works is cited or compared against in the paper. SOFTS and TimesXer, both cited in the paper, are also absent as baselines. The combination of components is well motivated, but the distinction from the broader sparse graph literature remains unclear, leaving the novelty of the contribution insufficiently justified.

---

> ### Author Rebuttal · Authors · 2026-03-31
>
> We appreciate your valuable suggestions and provide our detailed responses below.
>
> W1: Regarding the performance on the Traffic dataset, we provide the following clarification: the Traffic dataset exhibits stronger local spatio-temporal correlations. In scenarios with such dense and structurally stable dependencies, preserving richer interactions (e.g., TimeFilter) may offer an advantage. In contrast, MS-FLOW focuses on suppressing redundant interactions and noise propagation, making its strengths more apparent in datasets with high noise or redundancy. Although MS-FLOW ranks second on Traffic, it significantly outperforms baselines on more complex and heterogeneous datasets like Weather and Solar; thus, this result does not contradict our core thesis. Regarding the empirical support for "dense mixing leading to over-smoothing," we provided indirect evidence in Section 4.5: model performance degrades as interaction density increases. We also clarify that the "11 models" mentioned include MS-FLOW, while the 10 methods listed in Section 4.1 are the comparative baselines. We will clarify these points in the revised manuscript to complete the argument.
>
> W2: We appreciate the reviewer's positive comments on the paper’s structure and clarity, as well as the constructive suggestions. We will unify the notation in the revision: the input length will remain $L$, while the encoder depth will be denoted as Layers. The projection dimension $d_k$ in Eq. (8) corresponds to the embedding dimension in the graph learner, which is set to $64$. We will also resolve the discrepancy between "node_feat" in Figure 2 and $H$ in the text. Furthermore, we will improve the readability of the formulas in Figure 2 and fix the font embedding issues in the PDF. Regarding the Related Work, due to space constraints, the detailed discussion was placed in Appendix F; per your suggestion, we will move a clearer overview of related work into the main text, including discussions on GNN-based methods.
>
> W3: Thank you for this suggestion. Please refer to our response to W3 from Reviewer 1 (wjSP).
>
> W4: We thank the reviewer for the rigorous comments and for recommending related works such as Graph WaveNet, MTGNN, SOFTS, and TimeXer. We acknowledge that the individual operators in our model are not entirely new. However, our main contribution lies in reframing cross-variable interaction as a capacity-limited information flow problem. By combining patch representations with Top-K routing, MS-FLOW builds a unified sparse dependency bottleneck that suppresses redundancy, noise propagation, and over-smoothing. Compared with spatio-temporal GNNs, MS-FLOW also differs in both design philosophy and modeling granularity: existing methods usually learn graph structures directly on point-wise temporal dynamics, making them more sensitive to high-frequency noise, whereas MS-FLOW first extracts more stable patch-level contextual features and then applies the sparse bottleneck. In addition, we have added SOFTS and TimeXer to our main comparisons (Table 3 in the anonymous link), which we believe more fully validates the effectiveness and innovation of our approach.
>
> Q1 - Q3: We thank the reviewer for these comments. We have addressed them in the W1 and W3 sections of our response, respectively.
>
> Q4: Thank you for this suggestion. We analyzed the learned sparse graphs on the Weather dataset in Appendix H2. Specifically, we visualized the learned adjacency matrices and found that: in the early encoding stages, the dependency graph closely resembles the correlation structure within the input window, capturing state dependencies from observed history. As the encoding depth increases, the dependency graph is reshaped to align more closely with the correlation structure of the future sequence. This demonstrates that the learned sparse dependencies are not only beneficial for performance but also offer interpretability by reflecting underlying variable relationships.
>
> Q5: We appreciate this insightful question. In MS-FLOW, the Top-K edges are not static; they are recomputed at each forward pass based on the current input representation, making them sample-adaptive and evolving with the feature space. At the same time, we observe that the overall dependency patterns gradually become stable during training: strong and reliable variable relationships are consistently selected, while weak or noisy connections are suppressed. Therefore, the Top-K graph in MS-FLOW can be viewed as a dynamic yet stabilizing dependency structure, providing both adaptability and robust sparse routing. We will clarify this property in the revised manuscript.
>
> Once again, we sincerely thank you for your time and all your valuable comments. We hope that these additional experiments and detailed responses fully address your concerns, and we kindly ask you to reconsider your assessment of our work in the final evaluation.
>
> Anonymous link: https://anonymous.4open.science/r/ICML-4B7E

---

> > ### Author Rebuttal · Reviewer_Q1ew · 2026-04-01
> >
> > I thank the authors for the detailed and constructive rebuttal. Several of my concerns have been resolved. The explanation for the Traffic dataset is reasonable, the baseline count is clarified, the notation will be fixed in the revision, and the addition of SOFTS and TimeXer as baselines is appreciated. The answer regarding Top-K edge stability was also informative.
> >
> >
> > One point remains open for me. The over-smoothing claim is central to the motivation of the paper, but the evidence provided remains indirect. The observation that performance drops with denser interactions could have many explanations beyond over-smoothing. A direct measurement would be more convincing, for example comparing the similarity of variable representations under dense vs. sparse routing. If the representations converge under dense mixing but remain distinct under sparse routing, that would directly support the claim.

---

> > > ### Author Response · Authors · 2026-04-02
> > >
> > > We sincerely thank the reviewer for this exceptionally constructive and insightful suggestion.
> > >
> > > Following your advice, we conducted a direct empirical analysis by tracking and comparing the hidden representations of variables under dense routing (i.e., setting $K=C$, which allows full interaction among all variables) and our proposed sparse routing in MS-FLOW (i.e., imposing a Top-$K$ bottleneck). To ensure the robustness and comprehensiveness of this analysis, we quantified the over-smoothing effect across all encoder layers using two complementary metrics. Let $h_i \in \mathbb{R}^D$ denote the hidden representation of the i-th variable, and let $C$ denote the total number of variables. The metrics are defined as follows:
> > >
> > > 1.Average Cosine Similarity, which measures the directional alignment among variable representations:
> > >
> > > $$Cosine(H)=\frac{1}{C(C-1)}\sum_{i \neq j}\frac{h_i \cdot h_j}{\|h_i\|_2 \|h_j\|_2}$$
> > >
> > > 2.Average Pearson Correlation Coefficient, which measures the linear correlation and fluctuation similarity among variables, where $\bar{h}_i$ denotes the mean of vector $h_i$:
> > >
> > > $$Pearson(H)=\frac{1}{C(C-1)}\sum_{i \neq j}\frac{(h_i-\bar{h}_i)\cdot(h_j-\bar{h}_j)}{\|h_i-\bar{h}_i\|_2 \|h_j-\bar{h}_j\|_2}$$
> > >
> > > We carried out this empirical study on five benchmark datasets, including highly heterogeneous datasets (Weather and Electricity) as well as more spatially homogeneous traffic datasets (PEMS03, PEMS04, and PEMS08). The results are shown in Figure 2-4 (Anonymous link: https://anonymous.4open.science/r/ICML-4B7E).
> > >
> > > Across Weather, Electricity, PEMS03, PEMS04, and PEMS08, the dense routing consistently yields higher average cosine similarity and higher average Pearson correlation than sparse routing at all encoder layers. Moreover, in most cases, both curves continue to increase as the network goes deeper. This directly indicates that when all variables are allowed to interact without restriction, their hidden representations are more likely to collapse toward similar directions. In contrast, under the Top-K sparse bottleneck, the model still performs necessary information fusion, while preserving substantially stronger representational diversity across variables. In other words, these results provide direct representation-level evidence that dense interaction is more prone to inducing representation homogenization, whereas sparse routing mitigates this tendency.
> > >
> > >
> > > A closer look further reveals that this phenomenon is consistent across datasets, although its intensity varies. On Electricity, which is a more heterogeneous multivariate setting, the separation between dense and sparse routing is more pronounced. This suggests that when variables differ more substantially in semantics and statistical properties, unconstrained full mixing is more likely to pull their representations together, thereby causing stronger homogenization. By contrast, on the PEMS datasets, where variables are naturally more homogeneous due to the spatial structure of traffic networks, the similarity values under both routing strategies are already relatively high. Nevertheless, dense routing remains systematically higher than sparse routing, showing that even in settings with inherently strong correlations, the sparse bottleneck still suppresses further representational collapse rather than merely weakening interaction capacity.
> > >
> > > More importantly, these results help disentangle performance degradation from over-smoothing. A drop in forecasting accuracy under denser interactions could indeed admit multiple explanations. However, by directly measuring inter-variable representation similarity across layers, we now observe that dense routing not only leads to worse predictive performance, but also consistently produces higher representational similarity. Therefore, our conclusion no longer relies solely on indirect evidence. Instead, we can more directly state that the observed performance degradation is accompanied by a measurable convergence of variable representations, which is fully consistent with the mechanism of over-smoothing. Put differently, the advantage of our method lies not merely in “removing some edges,” but in restricting the bandwidth of cross-variable information flow, thereby preventing all variables from being excessively averaged together through repeated propagation.
> > >
> > > We thank the reviewer again for the invaluable suggestion and the time invested in reviewing our manuscript. We hope that our detailed response and the additional experimental evidence have satisfactorily addressed this concern. We kindly ask the reviewer to reassess our work in light of these substantial revisions.

---

### Official Review · Reviewer_UoyU · 2026-03-09

**Soundness:** 4
**Presentation:** 3
**Significance:** 3
**Originality:** 3
**Overall Recommendation:** 5
**Confidence:** 4

**Summary:**

This paper investigates the problem of cross-variable dependency modeling in multivariate time series forecasting. The authors point out that existing methods typically employ channel-independent modeling or dense interaction mechanisms (such as attention or channel mixing). The former may overlook potential variable interactions, while the latter is prone to propagating redundancy or spurious correlations. To address this, this paper proposes the MS-FLOW framework, which models cross-variable interactions as a capacity-constrained information flow and introduces a sparse dependency bottleneck to selectively control information transmission between variables. The model first divides the input sequence into overlapping time blocks and embeds them using a time block encoding module to extract local temporal features. Then, it uses a block-level time interaction module to model the temporal dependencies within variables. Finally, it learns an adaptive cross-variable dependency structure through the sparse dependency bottleneck. Specifically, a similarity matrix is calculated using query and key representations, and a Top-K sparsification strategy is employed to ensure that each variable interacts only with a few of the most relevant variables. The resulting sparse dependency graph is shared across different time blocks, and cross-variable information propagation is achieved through graph aggregation. Experimental results show that this method outperforms several state-of-the-art baseline models on multiple benchmark datasets.

**Compliance With Llm Reviewing Policy:**

Affirmed.

**Final Justification:**

Thank you for the clear and detailed rebuttal. My main concerns have been addressed. I am more confident in the work and am happy to increase my score.

**Key Questions For Authors:**

Please refer to the Weaknesses section under "Strengths and Weaknesses".

**Limitations:**

Yes

**Strengths And Weaknesses:**

Strengths：

S1: This paper addresses a crucial challenge in multivariate time series forecasting: modeling inter-variable dependencies while minimizing redundant interactions.

S2: The proposed sparse dependency bottleneck provides a clear and intuitive mechanism for restricting information flow between variables.

S3: This method combines block-based temporal modeling and sparse intervariable interactions within a coherent framework.

S4: Experiments on multiple benchmark datasets demonstrate that this method offers continuous improvements over robust baseline methods, and ablation experiments validate the effectiveness of the proposed module.

Weaknesses

W1: The learned sparse dependency matrix is shared across all image patches within the window. Could the authors explore whether the dependency structure for a specific image patch could further improve the flexibility of modeling?

W2: Why use a Top-K graph instead of a sparse attention mechanism?

W3: The paper lacks further analysis of the learned sparse variable dependency structure, such as whether these dependencies are interpretable or reflect the relationships between the true variables.

---

> ### Author Rebuttal · Authors · 2026-03-31
>
> We appreciate your valuable suggestions and provide our detailed responses below.
>
> W1: Thank you for your comment. Modeling patch-specific dependency structures could indeed provide greater flexibility. However, such a design would introduce substantial computational overhead. Specifically, modeling dependencies at the patch level requires interactions across both the channel and patch dimensions, leading to a much denser computation pattern. In contrast, MS-FLOW adopts a shared dependency structure across patches, which effectively reduces the computational cost while maintaining strong modeling capability. Moreover, we have empirically compared this design with a patch-specific dependency modeling method (TimeFilter). As shown in Fig. 4, our method achieves competitive or superior performance while being significantly more efficient.
>
> W2: Sparse attention is a commonly used and powerful tool in sequence modeling. However, the attention matrix is dynamically computed from query-key interactions at each step. In multivariate time series, local perturbations or high-frequency noise spikes (as illustrated in Fig. 1) can easily hijack these dynamic attention weights, leading to unstable routing and spurious correlations. In contrast, our Top-K graph learns a stable routing structure within a specific look-back window. This stable topology acts as a robust regularizer, preventing high-frequency local anomalies from dynamically altering key communication paths. We further conducted a dedicated comparison, as shown in the table below, to explicitly compare our Top-K graph design with sparse attention mechanisms. The results show that MS-FLOW achieves the best performance on all datasets, indicating that imposing structured sparsity along the variable dimension is more effective and stable than sparsifying attention over the token dimension. At the same time, the results of Dense Attention and Dense Graph further verify that excessive interactions introduce redundant dependencies and even noise propagation, thereby weakening forecasting performance.
> | Model Variant      | ETTh1 | ETTm1 | Weather | Traffic |
> |------------------|------:|------:|--------:|--------:|
> | **MS-FLOW (Ours)** | **0.367** | **0.302** | **0.150** | **0.407** |
> | Sparse Attention  | 0.381 | 0.326 | 0.161 | 0.432 |
> | Dense Attention   | 0.376 | 0.331 | 0.167 | 0.421 |
> | Dense Graph       | 0.377 | 0.316 | 0.158 | 0.418 |
>
> W3: We thank the reviewer for this valuable suggestion. The analysis of the learned sparse dependency structures is already included in the original submission (see Appendix H2). In that section, we visualize the learned adjacency matrices. At the early stage of encoding, the learned dependency graph is highly similar to the correlation structure within the input window, indicating that the model initially captures the state-dependent couplings present in the observed history. As the encoding depth increases, the dependency graph is gradually reshaped and becomes more aligned with the correlation structure of the future sequence. These observations suggest that the learned sparse dependencies are not only beneficial for prediction, but also exhibit a degree of interpretability and consistency with underlying variable relationships. We will further emphasize this part in the main text to make the analysis clearer.
>
> Once again, we sincerely thank you for your time and all your valuable comments. We hope that these additional experiments and detailed responses will fully address your concerns, and we kindly ask you to reconsider your assessment of our work in the final evaluation.

---

> > ### Author Rebuttal · Reviewer_UoyU · 2026-04-03
> >
> > I thank the authors for their detailed and constructive rebuttal. Most of my previous concerns have now been addressed. However, I still have one question: do the variables in these datasets exhibit clear intrinsic correlations? It would be very convincing if the authors could demonstrate this by visualizing the Pearson correlation matrices of the raw data. If this issue is properly addressed, I would be willing to raise my score.

---

> > > ### Author Response · Authors · 2026-04-03
> > >
> > > We thank the reviewer for the recognition and the valuable suggestion.
> > >
> > > We fully agree that directly showing the intrinsic correlations among variables from the raw data would provide more convincing evidence for the necessity of cross-variable modeling. Following your suggestion, we have added a visualization of the Pearson correlation matrices of the raw data in Figure 5 (Anonymous link: https://anonymous.4open.science/r/ICML-4B7E).
> > >
> > > As shown in the figure, clear correlation structures exist across multiple datasets, indicating that the variables are not simply independent channels. In particular, more pronounced cross-variable correlation patterns can be observed in datasets such as Traffic, PEMS, and Electricity, suggesting that these scenarios indeed contain strong cross-variable dependencies and are therefore more suitable for explicit cross-variable modeling methods.
> > >
> > > In contrast, the ETT datasets contain fewer variables and exhibit relatively simpler correlation structures, resulting in weaker cross-variable dependencies. We will incorporate this visualization and the corresponding discussion into the revised manuscript.
> > >
> > > We again thank the reviewer for this suggestion, as it has helped us present the data characteristics relevant to our method more clearly.

---

### Official Review · Reviewer_wjSP · 2026-03-13

**Soundness:** 3
**Presentation:** 3
**Significance:** 3
**Originality:** 2
**Overall Recommendation:** 4
**Confidence:** 4

**Summary:**

This paper proposes a sparse bottleneck framework MS-FLOW that explicitly models inter-variable interaction as capacity-limited information flow, which can preserve only a few informative inter-variable interaction while suppressing irrelevant ones. Experiments demonstrate the effectiveness of MS-FLOW.

**Compliance With Llm Reviewing Policy:**

Affirmed.

**Final Justification:**

This paper proposes a sparse bottleneck framework MS-FLOW to preserve a few informative inter-variable interactions while suppressing irrelevant ones. The paper is well-presented and easy to follow. Although the design of MS-FLOW appears to be largely built upon existing components, comprehensive experiments demonstrate its effectiveness and soundness. For studies on channel modeling strategies in multivariate time series forecasting, MS-FLOW provides a useful reference and a strong baseline. The detailed rebuttal addressed most of my concerns. The authors provide additional analysis suggesting that combining dynamic graph learning with sparsification strategy does not lead to performance gains. The inclusion of the recent state-of-the-art baselines also strengthens the validity of MS-FLOW. However, the design of MS-FLOW just combines existing components (i.e., sparse graph learning and patch modeling) rather than introducing technical novelty. Therefore, I still have concerns regarding the originality of the work. In addition, I appreciate the authors’ plan to explore the adaptive sparsity mechanism for automatically determining the number of effective dependencies. I look forward to future progress in this direction. Considering the authors’ rebuttal, I raise Significance score from 2 to 3 and raise my Overall Recommendation from 3 to 4.

**Key Questions For Authors:**

N/A

**Limitations:**

yes

**Strengths And Weaknesses:**

Strengths

1.The experiments are presented in considerable detail.

2.The paper is well written and easy to follow.

Weaknesses

1.The design of sparse bottleneck framework lacks innovation and appears to be a standard sparsification strategy, which is widely explored in prior works (e.g., MAGNN [1]). The authors should clarify the relationship between MS-FLOW and these works to further emphasize their own contribution. In addition, the approach of obtaining patch representation to resist short-term noise is also common in other papers (e.g., PatchTST [2] and Pathformer [3]).

2.The design of shared sparse dependency graph across all patch positions to learn dependency structure correspond to the window state is a little unreasonable. It presupposes that inter-variable dependencies are static within the look-back window, while these dependencies are often time-varying. For instance, in traffic flow forecasting, the correlation structures during rush hour is very different from that during off-peak hours.

3.The manual tuning of hyperparameter K for different datasets is a notable weakness. This requires costly empirical searching for new datasets and reduces the practical applicability of MS-FLOW. The authors are encouraged to analyze the relationship between the optimal K and the dataset properties deeply.

4.The paper has some weaknesses in the experiments, which are not convincing enough:

(1)Some recent state-of-the-art baselines should be compared to further validate the effectiveness of MS-FLOW, e.g., TimeEmb [4] and TimeKAN [5].

[1]Chen L, Chen D, Shang Z, et al. Multi-scale adaptive graph neural network for multivariate time series forecastin. TKDE, 2023.

[2]Nie Y, Nguyen N H, Sinthong P, et al. A time series is worth 64 words: Long-term forecasting with transformers. ICLR, 2023.

[3]Chen P, Zhang Y, Cheng Y, et al. Pathformer: Multi-scale transformers with adaptive pathways for time series forecasting. ICLR, 2024.

[4] Xia M, Zhang C, Zhang Z, et al. TimeEmb: A lightweight static-dynamic disentanglement framework for time series forecasting. NIPS, 2025.

[5] Huang S, Zhao Z, Li C, et al. TimeKAN: KAN-based frequency decomposition learning architecture for long-term time series forecasting. ICLR, 2025.

---

> ### Author Rebuttal · Authors · 2026-03-31
>
> We appreciate your valuable suggestions and provide our detailed responses below.
>
> W1: Although existing works such as MAGNN also explore sparsification strategies, the sparse dependency bottleneck in MS-FLOW differs fundamentally from them in terms of motivation, computational efficiency, and modeling granularity. We highlight the following three key differences:
>
> * Motivation: MAGNN first computes K dense node similarity matrices and then applies sparsification. This is essentially a computational compromise, aiming to prune an already dense network. In contrast, the sparse bottleneck in MS-FLOW is a proactive structural constraint mechanism designed to intentionally “forget” redundant information. It limits the communication budget from the outset, thereby physically preventing over-smoothing and the propagation of spurious correlations.
>
> * Computational efficiency: MAGNN needs to learn K independent dense adjacency matrices to capture multi-scale dependencies, leading to a time complexity of O(KN^2). By contrast, MS-FLOW learns only a single unified adjacency matrix with strictly constrained routing paths, which substantially reduces both computation and memory cost.
>
> * Granularity: Existing methods usually learn dependencies directly on raw variables at individual time steps, making them more vulnerable to high-frequency noise and local perturbations. In contrast, MS-FLOW first obtains stable representations through patch-level compression and then applies the sparse bottleneck, so that sparse routing is built on more robust patch representations rather than noise-sensitive point-level data.
>
> Regarding patch-based models such as PatchTST and Pathformer, we acknowledge that they also use patching techniques, but our motivation is fundamentally different. PatchTST is an early work introducing patching, and Pathformer further improves adaptive patching. Their main goal is to enhance temporal representation learning. In MS-FLOW, however, patch representations are not only used for temporal feature extraction, but more importantly serve as stable carriers for cross-variable structure learning, directly supporting the subsequent sparse dependency modeling.
>
> W2: We fully agree with your point that variable dependencies may change over time. As illustrated in Fig. 1(a), we are fully aware of this highly dynamic characteristic in real-world scenarios. However, we would like to clarify that sharing a graph within a specific look-back window is a deliberate design choice intended to balance “robust topology” and “temporal dynamics,” rather than assuming that dependencies are absolutely static. More importantly, although the correlation structure may vary significantly across different system states, the underlying driving pathways of the system remain inherently sparse. In our framework, the model uses a small K to handle such variation. We find that, regardless of the current system state, a small K is sufficient to capture the key cross-variable collaborative information. By enforcing a small K, the bottleneck actively filters out unstable edge correlations. It is precisely this mechanism that enables MS-FLOW to adapt robustly to diverse dynamic conditions. Our empirical results, shown in Figures 3 and 5, strongly demonstrate that even in highly dynamic datasets, maintaining tight sparse dependencies consistently delivers superior relevance modeling and prediction performance.
>
> W3: Thank you for raising this important question. We agree that the choice of K is related to dataset characteristics. However, our analysis (shown in Fig. 1 at the anonymous link) indicates that the cumulative dependency ratio rises rapidly with the first few neighbors, suggesting that most effective dependencies are concentrated on a small number of key variables rather than being uniformly distributed across all variables. Therefore, a simple and effective heuristic is to set K as a small fraction of the number of variables, e.g., K=13 for Electricity (C=321) and K=15 for Solar (C=137). This suggests that, in practice, it is unnecessary to conduct an extensive search over all possible values of K. We have already discussed this point in the limitations section of the original manuscript (Appendix H3). Following your suggestion, we will further explore adaptive sparsity mechanisms in future work to automatically determine the number of effective dependencies and reduce the need for manually setting K.
>
> W4: We have added comparisons with TimeEmb and TimeKAN. As shown in Table 3 at the anonymous link, MS-FLOW outperforms these methods on most benchmarks. We will include these baselines in the revised version.
>
> Once again, we sincerely thank you for your time and all your valuable comments. We hope that these additional experiments and detailed responses fully address your concerns, and we kindly ask you to reconsider your assessment of our work in the final evaluation.
>
> Anonymous link: https://anonymous.4open.science/r/ICML-4B7E

---

> > ### Author Rebuttal · Reviewer_wjSP · 2026-04-02
> >
> > Thank you to the authors for the detailed rebuttal. My concerns are partially addressed. However, I still have the following concerns. First, the sparsification strategy in MAGNN is not only motivated by computational efficiency, but also aims to improve robustness and reduce noises, which is conceptually similar to the motivation of MS-FLOW in preventing over-smoothing and spurious correlations. This concern has also been raised by other reviewers (e.g., Reviewer Q1ew). Second, the sparsification strategy and dynamic dependencies are not mutually exclusive. The authors are encouraged to include a variant with dynamical graph learning and a sparsification strategy to evaluate this design.

---

> > > ### Author Response · Authors · 2026-04-03
> > >
> > > Q1: We thank the reviewer for this important comment. MAGNN is a highly relevant and insightful work, and its perspective of modeling multiscale dependencies provides valuable guidance for multivariate time series forecasting. We also agree with the reviewer that both MAGNN and our method employ sparsification, and thus share the high-level intuition that fully dense interactions may not be optimal. However, the two methods differ substantially in modeling granularity and computational structure. To better highlight the novelty of our method, we provide a comprehensive comparison along several key dimensions in the table below.
> > >
> > > | Aspect                 | MS-FLOW                | MAGNN                          |
> > > |------------------------|------------------------|---------------------------------|
> > > | Number of learned graphs | 1                    | S                               |
> > > | Complexity             | O(C²)                  | O(SC²)                          |
> > > | Graph sharing scope    | Shared across all patches | Shared within each scale     |
> > > | Main purpose           | Sparsify interactions  | Control interactions across scales |
> > >
> > > MAGNN learns multiple scale-specific graphs at the channel level, with complexity $O(SC^2)$, where $S$ denotes the number of scales and $C$ the number of variables. In contrast, MS-FLOW learns only a single sparse dependency graph after patch compression, with complexity $O(C^2)$, and this graph is shared across all patches.
> > >
> > > Conceptually, MAGNN and MS-FLOW address different problems. The main motivation of MAGNN is multi-scale dependency modeling: it assumes that temporal patterns at different scales should be associated with different inter-variable graphs. In contrast, the motivation of MS-FLOW is to control cross-variable information flow: it treats over-interaction as the core failure mode and uses a sparse bottleneck to suppress noise, redundancy, and spurious dependencies. Therefore, although both methods involve sparsification at a high level, MAGNN uses it as part of multi-scale graph construction, whereas MS-FLOW elevates it to the core principle for robust interaction control. This leads to fundamentally different modeling behaviors: MAGNN attempts to identify important edges, while MS-FLOW explicitly limits the total amount of communication allowed between variables. In other words, MAGNN answers the question of “which connections are important,” whereas MS-FLOW answers “how much information should be allowed to flow across variables.” This distinction in modeling perspective is the key difference between the two methods.
> > >
> > > This difference also leads to different empirical outcomes. As shown in the table below, MS-FLOW consistently outperforms MAGNN on six public datasets. These results suggest that MS-FLOW is not simply a variant of MAGNN-style sparsification, but rather a distinct framework that improves robustness by enforcing selective interactions.
> > >
> > > | Dataset     | MS-FLOW | MAGNN |
> > > |-------------|---------|-------|
> > > | ETTh1       | 0.367   | 0.382 |
> > > | ETTm1       | 0.302   | 0.321 |
> > > | Weather     | 0.150   | 0.165 |
> > > | Solar       | 0.173   | 0.200 |
> > > | Electricity | 0.132   | 0.153 |
> > > | Traffic     | 0.407   | 0.431 |
> > >
> > > Q2: We thank the reviewer for this suggestion. We agree that sparsification and dynamic dependency modeling are not mutually exclusive, and their combination is indeed a valuable direction. In fact, we have already provided an ablation study of such a variant in Table 2 (Anonymous link: https://anonymous.4open.science/r/ICML-4B7E), where we introduced a design that combines both dynamic graph learning and sparsification. The results show that, although this variant offers stronger modeling flexibility, its performance does not consistently surpass the current shared sparse graph design, while also incurring higher computational cost and greater training instability. This further suggests that the current window-level shared sparse dependency structure achieves a better balance between forecasting performance and computational efficiency.
> > >
> > > We sincerely thank the reviewer once again for the helpful and constructive comments. We hope that the revised manuscript, together with the additional experiments and clarifications, has sufficiently addressed the concerns. We would greatly appreciate the reviewer’s reconsideration of the score and overall assessment in the final round.

---

### Decision · Program_Chairs · 2026-04-30

**Decision:**

Accept (regular)

**Comment:**

The paper proposes MS-FLOW, a sparse-bottleneck framework that models cross-variable dependencies in multivariate time series forecasting by enforcing capacity-limited information flow through selective sparse routing.

While the paper has some limitations, reviewers generally found the work to be technically sound and empirically effective after the constructive rebuttal responses by the authors. First, the proposed framework is well-motivated, addressing the important issue of spurious correlations and over-interaction in multivariate forecasting. The sparse dependency bottleneck provides a clear and intuitive mechanism, and the overall design is coherent. Second, the empirical evaluation is comprehensive, covering a wide range of datasets and demonstrating consistent improvements over strong baselines. Additional analyses provided in the rebuttal, such as direct evidence for over-smoothing and expanded baseline comparisons, further strengthen the credibility of the results and address key reviewer concerns. Third, the work is well presented, and the rebuttal successfully clarified most issues raised during the review process, leading to improved reviewers’ confidence.

That said, some concerns remain. In particular, the technical novelty is somewhat limited, as the method builds upon existing components such as sparse graph learning and patch-based representations. There are also practical limitations, including notation inconsistencies (L used for both input length and encoder depth, d_k unspecified, node_feat vs. H mismatch), missing comparison with spatiotemporal methods, manual tuning of K, and the used evaluation datasets.

Overall, the strengths in technical soundness, empirical validation, and potential impact outweigh these limitations. Therefore, this paper is recommended as weak accept.